# VORTA: Efficient Video Diffusion via Routing Sparse Attention

**Wenhao Sun**    **Rong-Cheng Tu**[§]  **Yifu Ding**    **Zhao Jin**
**Jingyi Liao**    **Shunyu Liu**    **Dacheng Tao**[§]

College of Computing and Data Science, Nanyang Technological University, Singapore

{wenhao006, rongcheng.tu, n2409547h, zhao.jin}@ntu.edu.sg
{jingyi012, shunyu.liu, dacheng.tao}@ntu.edu.sg

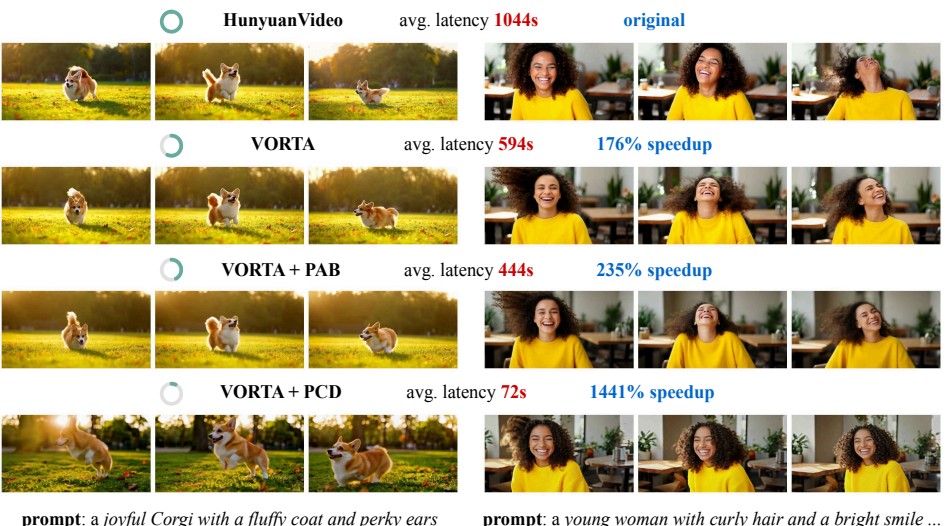

**prompt**: a *joyful Corgi with a fluffy coat and perky ears bounds through a sunlit park ...*

**prompt**: a *young woman with curly hair and a bright smile ... wearing a yellow sweater ...*

Figure 1: **VORTA** enables lossless acceleration of video diffusion transformers [16, 43], and remains compatible with other acceleration methods such as PAB [60] and PCD [44] for additional speedups.

## Abstract

Video diffusion transformers have achieved remarkable progress in high-quality video generation, but remain computationally expensive due to the quadratic complexity of attention over high-dimensional video sequences. Recent acceleration methods enhance the efficiency by exploiting the local sparsity of attention scores; yet they often struggle with accelerating the long-range computation. To address this problem, we propose **VORTA**, an acceleration framework with two novel components: (1) a sparse attention mechanism that efficiently captures long-range dependencies, and (2) a routing strategy that adaptively replaces full 3D attention with specialized sparse attention variants. VORTA achieves an end-to-end speedup $1.76\times$ without loss of quality on VBench. Furthermore, it can seamlessly integrate with various other acceleration methods, such as model caching and step distillation, reaching up to speedup $14.41\times$ with negligible performance degradation. VORTA demonstrates its efficiency and enhances the practicality of video

---

[§] Co-corresponding authors

39th Conference on Neural Information Processing Systems (NeurIPS 2025).

# 1 Introduction

Video Diffusion Transformers (VDiTs) have demonstrated impressive video generation performance, producing realistic and dynamic content [16, 33, 43, 51]. Despite this progress, VDiTs remain computationally expensive due to the inherently high-dimensional nature of video data, compounded by the quadratic complexity of attention operations. For example, recent HunyuanVideo [16] requires almost 1000 seconds (500 PFLOPS) to generate a 5-second 720p video at 24 frames per second (FPS) on an H100 GPU. To mitigate the high sampling cost, few-step sampling methods [39, 44, 52] leverage distillation on the self-consistency property of the probability flow ODE (PF-ODE) [38], achieving up to an $8\times$ reduction in sampling steps [56]. Other research [22, 26, 60] introduces intermediate feature caching to accelerate sampling without additional training.

Another promising direction investigates mitigating the quadratic computational complexity of the self-attention operation by leveraging its inherent sparse structure. The attention score distribution in VDiTs exhibits a dichotomy: local attentions and long-range attention. Local attentions [1, 53] tend to concentrate attention scores on a small subset of local keys around each query. Specifically, we observed that the spatially closest $4\%$ of the keys (4,000 out of 108,000) contribute more than $80\%$ of the total attention score as shown in Figure 2 (left). The remaining distant $96\%$ keys, which dominate the computational cost, contribute less than $20\%$. Approaches [1, 50, 53, 55] to mitigate this inefficiency in attention with local behavior restrict interactions to nearby tokens and discard distant ones. The second regime, long-range attention, distributes interactions across the entire sequence to capture global context and long-range dependencies. In these attention heads,

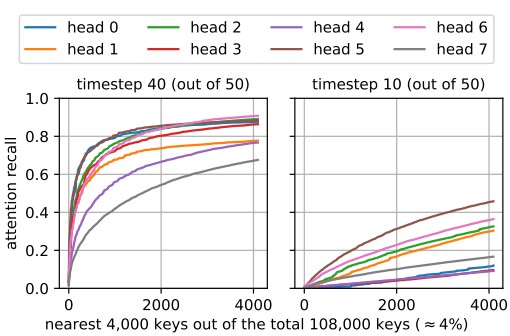

Figure 2: Attention scores recalled by the nearest keys. (*left*) Attention scores are predominantly concentrated within a local neighborhood. (*right*) The locality is less pronounced at earlier sampling steps. Results are from the 20th (of 60) layer in HunyuanVideo [16]. Only 8 out of the 24 attention heads are shown for clarity.

the closest keys account for less than $40\%$ of the total attention score as shown in Figure 2 (right). Directly imposing local attention constraint on these inherently nonlocal heads for acceleration compromises the critical global context modeling necessary for effective video generation [51]. Many works [47, 55] resort to dense operation in nonlocal heads, which severely limits the overall acceleration gain. Alternative unstructured or semi-structured attention pruning methods [40, 46, 54] also struggle with quadratic-time sparsity detection or profiling, introducing substantial overhead during the forward pass. Therefore, effectively and efficiently accelerating the long-range attention components in VDiTs remains a critical, unresolved challenge.

We observed that components exhibiting long-range dependencies are associated with high intra-sequence redundancy, whose tokens are highly similar. In this scenario, interactions with such redundant tokens are inefficient, as a few representative tokens can summarize their information. An example is provided in Figure 3 (middle), which depicts an intermediate generation result after the first few sampling steps. While high-level semantic structures, like spatial layouts and object motions, are primarily formed, the tokens have not acquired sufficient distinctiveness (*i.e.* highly redundant). This motivates our design of a token-level sparse attention

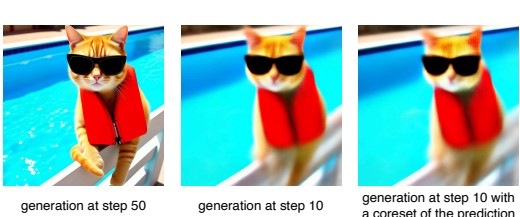

generation at step 50     generation at step 10     generation at step 10 with a coreset of the prediction

Figure 3: (*left*) Generation with the complete sampling process. (*middle*) Intermediate generation result. (*right*) Intermediate generation result with only core-set predictions.

to substitute the full-sequence computation when a compact core-set is sufficient to capture all necessary information. We propose a bucketed core-set selection (BCS) strategy to remove similar, redundant tokens and retain only the most representative ones for attention computation. As illustrated in Figure 3 (right), the output generated with core-set tokens retains semantic accuracy. Given that the core-set contains fewer tokens (*e.g.*, 50% of the full sequence), the attention cost is reduced quadratically (*e.g.*, to 25% of the original cost of the full sequence).

The implicit relationship between the VDiT's attention behaviors (*i.e.*, local or long-range dependencies) and the sampling steps, which correlates with the input signal strength, presents a challenge for the principled integration of sparse attention variants. Several approaches [47, 55] log attention score distributions through a reverse diffusion sampling process and reuse them. However, this is impractical in real-world settings where sampling configurations (*e.g.*, video size, sampling steps, or reverse diffusion solvers) frequently change, rendering the logged patterns obsolete and requiring re-tuning. To overcome this, we introduce a VORTA routing mechanism that dynamically selects the optimal sparse attention variant by leveraging the input signal-to-noise level. It provides precise control and adaptable support for diverse diffusion configurations, offering greater flexibility.

To conclude, our contributions are summarized as follows:

- We propose a novel core-set attention that effectively models long-range dependencies while theoretically reducing the attention cost by 75%.
- We introduce VORTA, an approach that integrates multiple sparse attentions into VDiTs for end-to-end acceleration without sacrificing performance. VORTA is compatible with various SDE/ODE solvers and network backbones.
- VORTA achieves a $1.76\times$ end-to-end speedup on VBench [11]. It is also compatible with other acceleration techniques, achieving overall speedups of up to $2.35\times$ with feature caching [60] and $14.41\times$ with consistency distillation [44].

## 2   Preliminary and related work

**Flow matching and diffusion models.**   Flow Matching (FM) [21, 23] builds upon Continuous Normalizing Flows (CNFs) [5] and has since been integrated with the diffusion paradigm [10, 38], forming the foundation of many recent diffusion models [9, 12, 16, 32, 43, 61]. The core concept of FM involves a time-dependent velocity field (VF) $v_t$ and a time-dependent diffeomorphism $\mathbf{x}_t := t\mathbf{x}_1 + (1-t)\mathbf{x}_0$, known as a *flow*. The VF governs the evolution of the flow through an ordinary differential equation (ODE) $d\mathbf{x}_t = v_t(\mathbf{x}_t)\,dt$, where the flow $\mathbf{x}_t$ defines a probability density path $p_t$, starting from the simple prior $p_0 = \mathcal{N}(\mathbf{0}, \mathbf{I})$ and evolving towards the intractable target density $p_1$.

Lipman et al. [21] introduced Conditional Flow Matching (CFM) to optimize the neural network $v_t^\theta(\mathbf{x}_t)$ (*i.e.* VDiTs in this context) as follows:

$$\mathcal{L}_{\text{CFM}} = \mathbb{E}_{t, p_1(\mathbf{x}_1), p_0(\mathbf{x}_0)} \| v_t^\theta(\mathbf{x}_t) - (\mathbf{x}_1 - \mathbf{x}_0) \|^2. \tag{1}$$

Once optimized, it can start from $\mathbf{x}_0 \sim p_0$ and follow the ODE trajectory to generate samples.

**3D self-attention of video diffusion transformers.**   Recent VDiTs [16, 18, 19, 33, 43, 51] employ 3D attention to capture spatio-temporal dependencies in video data. Given a video with $F$ frames at a resolution of $H \times W$, the input is flattened into a sequence of length $L = F \times H \times W$, denoted as $\mathbf{H} \in \mathbb{R}^{L \times d}$, where each token is a $d$-dimensional vector. The self-attention is formulated as:

$$\text{attn}(\mathbf{H}) = \text{softmax}\left((\mathbf{H}\mathbf{W}_Q)(\mathbf{H}\mathbf{W}_K)^\top\right)(\mathbf{H}\mathbf{W}_V), \tag{2}$$

where $\mathbf{W}_Q, \mathbf{W}_K, \mathbf{W}_V \in \mathbb{R}^{d \times d}$ are the linear projection matrices. The attention operation is of complexity $\mathcal{O}(L^2 d) + \mathcal{O}(Ld^2)$. In the context of high-resolution video processing, where the embedding dimension $d$ is significantly smaller than the sequence length $L$, the overall complexity is dominated by the first term $\mathcal{O}(L^2 d)$, which scales unfavorably with sequence length. Despite employing techniques such as Variational Autoencoders (VAEs) [15, 34] and patchification [31], the sequence length $L$ still reaches up to 100K for a 5-second 720p video in HunyuanVideo [16]. Consequently, attention operations dominate the computational cost, accounting for over 75% of the computation cost. Optimizing the attention computation is crucial for the efficiency of VDiTs.

**Video diffusion acceleration.** General diffusion acceleration methods include step distillation [24, 28, 35, 36, 39, 44], which reduces the number of sampling steps to as few as 4 to 8 with sufficient tuning, and feature caching [14, 22, 26, 58, 60], which avoids redundant computation by reusing intermediate features. In the case of VDiTs, the long sequence length leads to high attention costs. As a result, many studies have focused on optimizing attention for long sequences. Some recent approaches [47, 55] apply predefined sparse attention patterns to reduce computational overhead. Others [40, 46, 54] adopt online profiling to dynamically select attention patterns during inference. However, these sparse attention techniques depend on careful hyperparameter tuning for different configurations and/or introduce additional forward-pass complexity that offsets their speed benefits. VORTA addresses these challenges with a flexible design that is compatible with various diffusion configurations and model backbones, without incurring any additional inference-time overhead.

**Acceleration by conditional computing.** The concept of conditional computation has emerged as a powerful paradigm to accelerate neural networks by selectively activating only a subset of the parameters or operations for a given input. Bengio et al. [3] first introduced stochastic neurons in which parameter activation is conditional on their output. Subsequent methods selectively drop or gate layers [2, 27, 29, 37, 41, 45, 49], effectively utilizing a different, sparser network structure for each sample. In contrast to these prior efforts, our proposed VORTA integrates operation- and token-level sparsity and uses diffusion temporal dynamics to adaptively route computation.

## 3 VORTA: efficient video diffusion via routing sparse attention

This section introduces **VORTA** to accelerate VDiTs. We begin with a taxonomy of attentions in Section 3.1. Next, we present two core components: i) sparse attention variants that speed up specific attentions in Section 3.2; ii) a routing strategy that integrates these sparse attentions into pretrained VDiTs for end-to-end acceleration (Section 3.3).

### 3.1 Taxonomy of VDiT attentions

The attention sparsity has been briefly discussed in Section 1. We formally categorize attentions in VDiTs into three types:

- **Local attentions** focus on short-range interactions without attending to distant tokens. They are primarily responsible for fine-grained details during generation.
- **Long-range attentions** distribute their attention scores across the entire sequence. These attentions mainly capture high-level semantic information, including coarse layout and motion. Minor perturbations, such as merging similar tokens, are acceptable.
- **Pivotal attentions** maintain a global perceptual field while simultaneously refining local details. Small perturbations to these attentions can result in noticeable quality degradation.

All three types of attention coexist in VDiTs and are not mutually exclusive during the sampling: local attention may transition into nonlocal attention as the diffusion process evolves, and vice versa. We will provide supporting evidence for this taxonomy through experimental results in Figure 9.

### 3.2 Sparse attentions

**Sliding window for local attentions.** Sliding window attention [1, 53] proposes restricting each query token at position $i$ to attend to its local neighborhood within the range $(i-w, i+w]$, where $w$ is the window size. It provides an efficient alternative when the attention distribution is highly localized. However, its zigzag-shaped attention mask introduces computation bubbles in block-wise kernel implementations (*e.g.*, FlashAttention [6, 7]). This problem becomes more severe in three-dimensional video data, where the number of computation bubbles

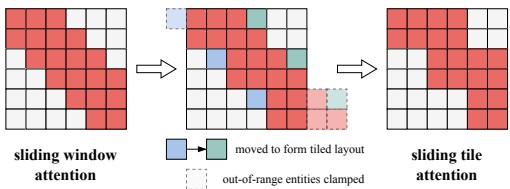

Figure 4: Illustration of converting a sliding window mask into a sliding tile mask. A 1D attention mask is shown for simplicity, with both the window size and tile size set to 2.

grows cubically. Zhang et al. [55] proposed sliding tile attention to tackle this problem. The 1D atten-

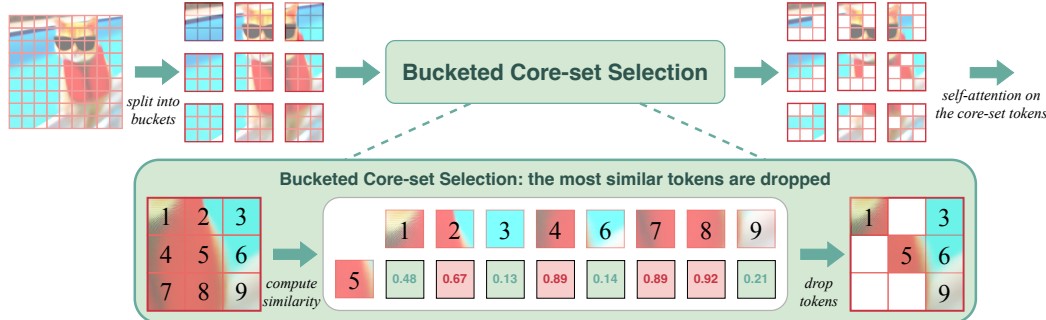

Figure 5: **Bucketed Core-set Selection (BCS).** For clarity, the 2D image is used for illustration; the actual inputs and buckets operate on 3D video data within the latent space. In this example, the top $k = 4$ tokens from each bucket are dropped.

tion mask is defined as: $\mathbf{M} = \{m_{i,j}\} = \{j \in (\tau(i) - w, \tau(i) + w]\}$, where $\tau(i) = \lfloor i/t \rfloor \cdot t + \lceil t/2 \rceil$ is the tile center, and $t$ is the tile size. It can be interpreted as shifting the window of non-center queries to align with the query in the center of the tile, as illustrated in Figure 4. The attention mask for sliding tile attention is block-wise dense and offers greater hardware efficiency. Our implementation employs 3D sliding tile attention to model local interactions of quality attentions with detailed pseudo code in Appendix A.1.

**Core-set selection for long-range attentions.** It is observed that many attentions with large perceptual fields are less sensitive to perturbations [40], corresponding to the long-range attentions defined earlier. Based on this inherent token-level sparsity, we introduce core-set attention to drastically accelerate computation by actively pruning redundant tokens during the attention operation. A prototype core-set attention operation is defined as follows:

$$\text{coreset-attn}(\mathbf{H}) = \text{unpool} \circ \text{attn} \circ \text{pool}(\mathbf{H}), \tag{3}$$

where $\circ$ denotes the composition operator, meaning the connected operators are applied sequentially from right to left. $\text{pool}(\cdot)$ and $\text{unpool}(\cdot)$ operations compress a long sequence into a compact core-set and recover the original sequence length, respectively.

Direct application of standard average pooling often results in a significant degradation of generation quality (see Section 4.2), because its core assumption, that all tokens within the pooling kernel are sufficiently similar, is often violated. As an alternative, we implement the $\text{pool}(\cdot)$ operation using Bucketed Core-set Selection (BCS), as outlined in Figure 5. It introduces a novel, buketed approach to achieve token-level sparsity. BCS first divides the tokens into buckets. Intra-bucket similarity is computed exclusively between a designated center token (token '5' in this example) and its neighboring tokens. Tokens with the top-$k$ highest similarity to the center are then pruned, and their representational information is merged into the center. Crucially, by eliminating inter-bucket similarity calculations, BCS achieves linear complexity $\mathcal{O}(L)$ for sequence length $L$, offering a significant computational advantage over $\mathcal{O}(L^2)$ sequence length pruning methods [4, 40] while maintaining competitive performance. A detailed complexity analysis is provided in Appendix A.1.

After applying the attention operation on the core-set tokens, we retrieve the original sequence length by scattering the center token back to the pruned tokens for the subsequent operations.

## 3.3 Signal-aware attention routing and detection

**Attention router.** Accurately identifying the optimal sparse attention variant is the remaining technical hurdle. We observe that attention behavior correlates strongly with the signal-to-noise ratio (SNR) of input features in Figure 2. To address this, we introduce a router implemented as a linear layer with diffusion timestep embedding $\mathbf{T} \in \mathbb{R}^d$ as input, following a softmax activation to output the gate values $\boldsymbol{\alpha}^{(n)}$ for dynamic attention variant selection:

$$\boldsymbol{\alpha}^{(n)} = \text{softmax}(\mathbf{T}\mathbf{W}_R^{(n)}), \tag{4}$$

where $\mathbf{W}_R^{(n)} \in \mathbb{R}^{d \times 3}$ denotes the linear projection matrix of the $n$-th transformer block. The gate values $\boldsymbol{\alpha}^{(n)} = \left[\alpha_1^{(n)}, \alpha_2^{(n)}, \alpha_3^{(n)}\right]$ quantify the suitability of full, sliding-window, and core-set

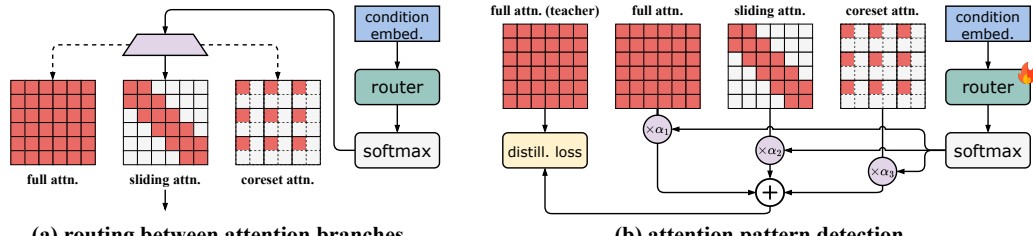

Figure 6: **Overview of VORTA.** 1D masks are used to represent the attention branches for simplicity. (*a*) During inference, the router takes the condition embedding as input and selects the appropriate attention branch. In this example, it activates sliding attention. (*b*) During pattern detection, all attention branches are activated, and their outputs are aggregated using gate values from the router. The divergence between the full attention output and the routed output is used to update the router.

attention operations, respectively. This lightweight design adds just 0.1% to the model's parameters, while enabling the router to adaptively modulate gate values based on the current signal-to-noise ratio (SNR), without directly operating on the full set of diffusion tokens.

**Inference-time routing strategy.**    The router activates the attention branch with the highest gate value, as illustrated in Figure 6 (a). It is formalized as a hard selection, defined as follows:

$$\mathbf{H}^{(n+1)} = \begin{cases} \text{sliding-attn}(\mathbf{H}^{(n)}) & \text{if } \alpha_2^{(n)} > \alpha_1^{(n)} \text{ and } \alpha_2^{(n)} > \alpha_3^{(n)} \\ \text{coreset-attn}(\mathbf{H}^{(n)}) & \text{if } \alpha_3^{(n)} > \alpha_1^{(n)} \text{ and } \alpha_3^{(n)} > \alpha_2^{(n)} \\ \text{attn}(\mathbf{H}^{(n)}) & \text{otherwise} \end{cases}, \tag{5}$$

where $\mathbf{H}^{(n)}$ represents the input features to the $n$-th transformer block. $\text{attn}(\cdot)$ is the standard full attention. $\text{sliding-attn}(\cdot)$ and $\text{coreset-attn}(\cdot)$ represent the sliding and core-set attentions, respectively, as defined in Section 3.2. Some operations, such as RoPE and FFN, have been omitted for clarity. Activating sparse attention branches yields a substantial speedup in attention computation. Although the full attention branch is activated in fewer than 0.2% of cases, it remains important for preserving model performance. We will empirically investigate these design choices in Section 4.2.

**Router optimization.**    Inspired by recent advances in language modeling [49], we adopt a self-supervised optimization strategy, as illustrated in Figure 6 (b). In the forward process, the gate values are employed to weight the outputs of the three branches:

$$\mathbf{H}^{(n+1)} = \alpha_1^{(n)} \cdot \text{attn}(\mathbf{H}^{(n)}) + \alpha_2^{(n)} \cdot \text{sliding-attn}(\mathbf{H}^{(n)}) + \alpha_3^{(n)} \cdot \text{coreset-attn}(\mathbf{H}^{(n)}). \tag{6}$$

We introduce a distillation loss $\mathcal{L}_{\text{distill}}$, defined as the MSE between the routed output $\mathbf{H}^{(N)}$ and the original output $\mathbf{H}_{\text{org}}^{(N)}$ of the final block, to ensure that the routed outputs remain close to the original model outputs to preserve the pretrained performance. In addition, we adopt the conditional flow matching loss $\mathcal{L}_{\text{CFM}}$ from the VDiT pretraining stage as detailed in Section 2. To promote sparsity in routing decisions, we apply an L2 regularization term to gate values. The final loss function combines the conditional flow matching loss, the distillation loss, and the regularization term, weighted by hyperparameters $\lambda_{\text{distill}}$ and $\lambda_{\text{reg}}$, respectively:

$$\mathcal{L} = \mathcal{L}_{\text{CFM}} + \lambda_{\text{distill}} \cdot \mathcal{L}_{\text{distill}} + \lambda_{\text{reg}} \cdot \mathcal{L}_{\text{reg}}, \tag{7}$$

$$\text{where } \mathcal{L}_{\text{distill}} = \text{MSE}\left(\mathbf{H}_{\text{org}}^{(N)}, \mathbf{H}^{(N)}\right) \text{ and } \mathcal{L}_{\text{reg}} = \sum_{n=1}^{N} \|\alpha_1^{(n)}\|^2. \tag{8}$$

Notably, the original parameters of the VDiTs are always frozen, and only the router is updated. We set $\lambda_{\text{distill}} = 20$ and $\lambda_{\text{reg}} = 0.02$ to balance the losses at a similar scale in our experiments. A detailed analysis of these hyperparameters is provided in Appendix B.1.

## 4    Experiments

**Baselines.**    This section presents the evaluation of the text-to-video generation task. We evaluate VORTA with two recently open-sourced VDiTs: HunyuanVideo [16] and Wan 2.1 [43]. Our

Table 1: Quantitative comparison under standard baseline settings (bf16, 720p, 5s). C : caching; D : step distillation; S : sparse attention.

| Models | Type | VBench ↑ | Quality ↑ | Semantic ↑ | LPIPS ↓ | Latency (s) | Speedup | Mem. (GB) |
|---|---|---|---|---|---|---|---|---|
| HunyuanVideo [16] | - | 82.26 | 83.68 | 76.60 | - | 1043.85 | 1.00× | 47.64 |
| + ARnR [40] | S | 82.39 | **83.85** | 76.56 | 0.211 | 790.55 | 1.32× | 78.15 |
| + STA [55] | S | 82.33 | 83.56 | 77.39 | 0.201 | 676.39 | 1.54× | 51.79 |
| + PAB [60] | C | 82.40 | 83.80 | 76.81 | 0.186 | 815.51 | 1.28× | > 80 |
| + VORTA | S | **82.59** | 83.74 | 77.95 | **0.185** | 594.23 | 1.76× | **51.15** |
| + VORTA & PAB | S & C | 82.56 | 83.60 | **78.38** | 0.195 | **444.19** | **2.35×** | > 80 |
| + PCD [44] | D | 81.17 | 82.56 | 75.35 | **0.564** | 125.98 | 8.29× | **47.64** |
| + VORTA & PCD | S & D | **81.49** | **82.78** | **76.31** | 0.575 | **72.46** | **14.41×** | 51.15 |
| Wan 2.1 (14B) [43] | - | 82.36 | 83.05 | 79.60 | - | 1304.82 | 1.00× | 41.77 |
| + VORTA | S | **82.85** | **83.45** | **80.44** | 0.222 | **856.50** | **1.52×** | 43.97 |

comparisons include sparse attention acceleration methods: STA [55], which adopts a predefined sparse attention pattern, and ARnR [40], which performs online profiling to determine the sparse attention pattern dynamically. We also compare VORTA against two orthogonal approaches: the caching-based method PAB [60] and the step distillation method PCD [44]. Additionally, we report results for VORTA combined with PAB and PCD, as these methods can be integrated. For the Wan 2.1 [43], we only compare the pretrained baseline and VORTA, since other methods have not yet released code to support this model.

**Benchmarks and evaluation metrics.** Following prior works [40, 60], we evaluate on the standard VBench prompt suite [11], which contains over 900 text prompts across 16 dimensions. The primary performance metric is the aggregated VBench score. We also report the VBench quality and semantic subscores to provide a more detailed breakdown. To assess the deviation of generation from the pretrained models, we use LPIPS [57] as a reference metric. For efficiency analysis, we measure video sampling latency of the VDiTs, excluding the time required for text encoding and VAE decoding. We report the relative speedup as $\Delta$latency/(latency + 1). Additionally, we record peak memory usage, which impacts practical deployment due to hardware constraints.

**Implementation.** We implement VORTA in PyTorch [30], using FlexAttention kernel for sliding attention and FlashAttention [6, 7] kernel for all other attention operations. Video samples are generated in 50 steps, except for PCD [44], which uses 6 steps. The videos are 5 seconds long at 720p resolution. Due to out-of-memory issues with PAB [60] at this resolution, we adopt sequential CPU offloading [42]. For fair comparison, the latency introduced by model loading and offloading is excluded in the experiment results. For router optimization, we use the Mixkit dataset [20], training for 100 steps with a learning rate of $10^{-2}$ and a batch size of 4. All experiments are conducted on H100 GPUs with 80GB of memory. Additional implementation details are provided in Appendix A.2.

## 4.1 Main results

**Performance.** Table 1 presents the quantitative evaluation results of the baseline methods and VORTA. The key observations are: i) All methods maintain high VBench scores, except PCD, which shows a 1-point drop due to aggressive compression of generation steps. ii) VORTA, with and without PAB, ranks first and second in VBench score and LPIPS on HunyuanVideo, respectively. iii) The good performance of VORTA on both the MMDiT-based [9] HunyuanVideo and DiT-based [31] Wan 2.1 demonstrates its generalizability across diffusion backbones. Figures 1 and 7 quantitatively depict the generations produced by ARnR [40], STA [55], VORTA, VORTA & PAB [60], and VORTA & PCD [44] on HunyuanVideo. All variants maintain high visual quality with vivid appearance and coherent motion, consistent with the quantitative metrics. More qualitative comparisons and complete VBench dimensional scores are provided in Appendix B.4 and Appendix B.5, respectively.

**Efficiency.** Among the sparse attention approaches, VORTA achieves the highest efficiency on HunyuanVideo, with a 1.76× speedup. For the other line of work, PAB achieves a 1.28× speedup on HunyuanVideo, but it requires over 1.7× additional memory, which makes it less practical for large models or high-resolution videos. In practice, running PAB on an 80GB GPU necessitates sequential

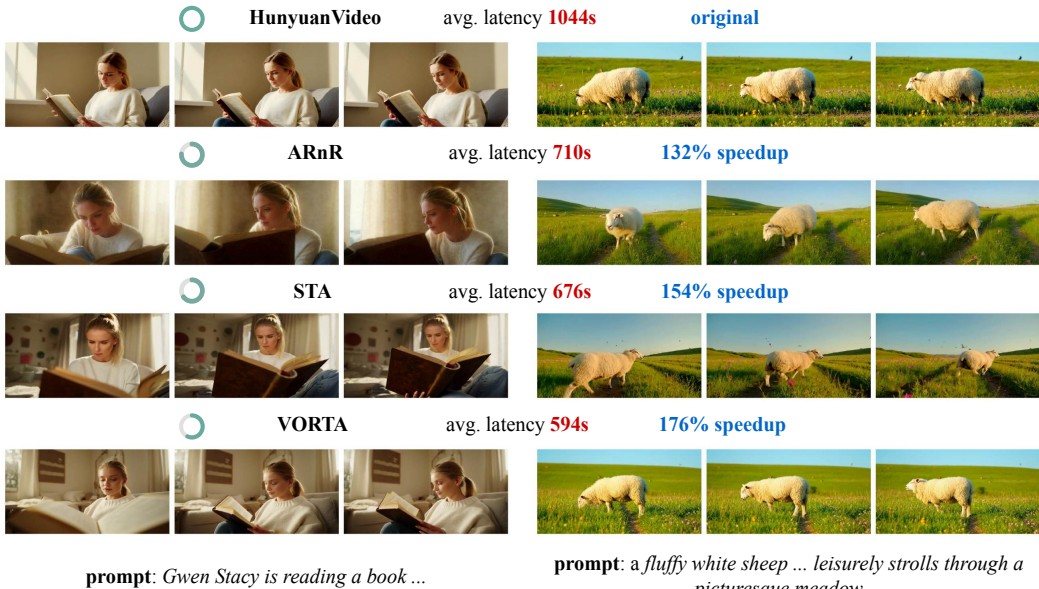

| | HunyuanVideo | avg. latency **1044s** | **original** |
| | ARnR | avg. latency **710s** | **132% speedup** |
| | STA | avg. latency **676s** | **154% speedup** |
| | VORTA | avg. latency **594s** | **176% speedup** |

**prompt**: *Gwen Stacy is reading a book ...*

**prompt**: a *fluffy white sheep ... leisurely strolls through a picturesque meadow ...*

Figure 7: Qualitative comparison of sparse attention methods: ARnR [40], STA [55], and VORTA.

**CPU offloading.** The latency introduced by parameter loading and offloading nearly offsets the benefits of computation. The step distillation approach, PCD, achieves an $8.29\times$ speedup using only 6 sampling steps, with a mild drop in performance. While these methods offer different trade-offs in efficiency and performance, both can be effectively integrated with VORTA. When combined with PAB and PCD, VORTA achieves total speedups of $2.35\times$ and $14.41\times$, respectively.

Table 2: Quantitative comparison on Wan 2.1 [43] with different sampling configurations.

| Models | Configuration | VBench | | | | Latency (s) | Mem. (GB) |
|---|---|---|---|---|---|---|---|
| | | Subject ↑ | Consistency ↑ | Aesthetic ↑ | Imaging ↑ | | |
| Wan 2.1 (14B) [43] | UniPC (50 step) [59] | 91.21 | **75.49** | 63.33 | **63.99** | 1359.76 | 41.77 |
| + VORTA | | 90.76 | 75.09 | 63.79 | 63.07 | 791.02 | 43.97 |
| Wan 2.1 (14B) [43] | DPM++ (30 step) [25] | 89.72 | 74.54 | **64.43** | **63.99** | 817.00 | 41.77 |
| + VORTA | | **91.82** | 75.13 | 64.32 | 63.95 | **440.76** | 43.97 |

**Generalization to various schedulers.** Unlike sparsity-inducing techniques that rely on offline profiling or hand-crafted heuristics to define a sparse execution strategy [40, 55], VORTA is designed to possess scheduler and step-count generalization, mirroring the flexibility of the baseline dense model. This obviates the need for costly re-profiling or hyperparameter re-tuning when changing sampling configurations. We demonstrate this capability by comparing the performance on the Wan 2.1 [43] using both the 50-step UniPC [59] and 30-step DPM++ [25] schedulers.

Table 2 presents the VBench dimensions [11] and efficiency metrics from these experiments, which were conducted on B200 GPUs. The consistent, lossless performance of VORTA relative to the dense model confirms its ability to generalize efficiently without added cost, an advantage enabled by its lightweight training, which heuristic-based methods like STA [55] and ARnR [40] do not achieve.

**Runtime breakdown.** Figure 8 further presents the average latency of individual components for the sparse attention methods. Here, "attn." denotes the isolated attention operation latency, whereas "attn. related" includes associated operations such as projections, RoPE, layer normalization, and other computations within the attention module.

Compared to ARnR [40], VORTA demonstrates superior efficiency. To preserve lossless generation, ARnR adopts a more conservative sparsity, resulting in higher latency in its "attn." component. Furthermore, its $\mathcal{O}(L^2)$ similarity computation introduces additional latency, as evidenced by the larger latency in its "attn. related" component. In contrast, VORTA adaptively identifies efficient sparse patterns and allows greater speedups. Its core-set attention also involves similarity computation; however, BCS has only $\mathcal{O}(L)$ complexity, incurring negligible additional cost.

VORTA also outperforms STA [55], which uses a predefined sliding attention pattern. The main bottleneck of STA occurs during the first 15 sampling steps (see bottom two bars), when it must expand the window size to capture long-range dependencies. Consequently, these steps do not benefit from attention sparsity. In contrast, VORTA adaptively routes attention branches to accommodate diverse interaction types while maintaining near-constant latency across all diffusion steps.

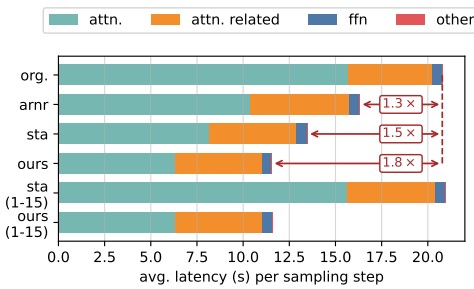

Figure 8: Runtime breakdown for sparse attention methods on HunyuanVideo [16].

Table 3: Evaluation of each VORTA component on Wan 2.1 (1.3B) [43].

| Models | VBench ↑ | Latency (s) | Speedup |
|---|---|---|---|
| Wan 2.1 (1.3B) | 81.20 | 73.24 | 1.00× |
| w/o Sliding Attn. | 80.25 | 65.14 | 1.12× |
| w/o Coreset Attn. | 79.89 | 66.10 | 1.11× |
| w/o Full Attn. | 77.14 | 59.34 | 1.23× |
| w/o Timestep Cond. | 81.03 | 65.00 | 1.13× |
| w/ $AP_{211}$ | 77.08 | **57.53** | **1.27×** |
| w/ $AP_{121}$ | 76.01 | 57.64 | **1.27×** |
| w/ $AP_{112}$ | 75.94 | 57.55 | **1.27×** |
| VORTA | **81.06** | 58.42 | 1.25× |

## 4.2 Ablation study

We conduct ablation studies on the Wan 2.1 (1.3B) [43] at its pretrained 480p resolution. All other experimental settings are identical to those used in the main experiment. At 480p, attention operation accounts for a smaller portion of the overall latency compared to 720p, resulting in less pronounced acceleration. Nevertheless, we adopt this smaller model and resolution to reduce the computational cost of benchmarking. The results are summarized in Table 3.

**Attention branches.**   We evaluate the contribution of the attention branches individually by removing them. When the sliding attention branch is disabled, the router shifts more selections toward the full attention expert to preserve performance, resulting in an over 10% runtime increase, despite applying the same regularization level. Similarly, removing the core-set attention branch increases the latency by $14\%$. These results highlight the importance of both sparse branches in improving efficiency through the adaptive selection of attention patterns tailored to specific characteristics.

Given that the router selects the full attention branch in only 0.2% cases in VORTA (will be detailed in Section 4.3), we assess whether retaining it is necessary. As shown in Table 3, the removal of the full attention branch leads to a 4-point drop in the VBench score without any additional speedup. This suggests that the full attention branch remains crucial and validates the existence of the pivotal attention patterns defined in Section 3.1.

**Timestep condition for signal-aware routing.**   We also examine the impact of including timestep information in attention routing. Removing the timestep embedding from the router input results in uniform attention branch selection across all diffusion steps, leading to slower 12%. In the absence of timestep conditioning, the router consistently selects the same attention expert. To maintain performance, the model predominantly routes to the full attention expert.

**Bucketed core-set selection (BCS).**   We evaluate how much BCS improves performance compared to conventional average pooling. BCS sets the core-set size to 50% of the original sequence length. To ensure a fair comparison, we apply the same reduction ratio in the average pooling cases by configuring the pooling kernel sizes such that their product equals 2. To isolate the effect of compression along different dimensions, we evaluate three pooling kernel configurations: $(2, 1, 1)$, $(1, 2, 1)$, and $(1, 1, 2)$, denoted as $AP_{211}$, $AP_{121}$, and $AP_{112}$, respectively. Although average pooling offers slightly lower latency by being free from similarity computations, its performance drops significantly. When adjacent tokens differ significantly, naive merging causes information loss, often resulting in artifacts such as pixelation or blurring in the generation.

### 4.3 Attention patterns in VDiTs.

Figure 9 illustrates the routed attention branch assigned to each head, layer, and sampling step. It reveals a clear temporal pattern: earlier time steps tend to use more corset attention, while later time steps increasingly rely on sliding attention. Only a small fraction (about $0.2\%$) of attention heads are assigned to the full attention branch, highlighting the sparsity. As intuitively expected, earlier time steps likely focus on constructing high-level semantics rather than fine-grained details, whereas later time steps emphasize local interactions. Unlike auto-regressive or discriminative models [8, 13], VDiTs exhibit weaker layer-wise specialization; attention heads within the same layer are more evenly distributed across branches. A minor tendency is observed in later time steps where sliding attention is more frequently assigned to intermediate layers, while corset attention is more often assigned to the initial and final layers. One plausible explanation is that intermediate layers capture local details, while subsequent layers refine the overall representation quality. Step-wise visualizations and results for other models are provided in Appendix B.3.

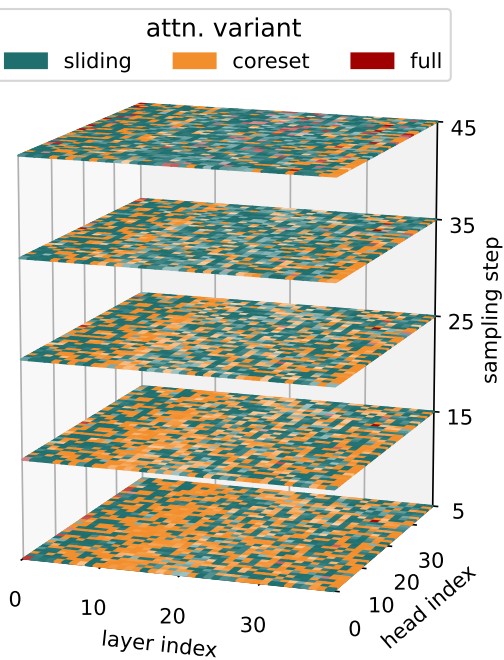

Figure 9: Optimized gate values for Wan 2.1 (14B) [43]. We visualize snapshots at sampling steps 5, 15, 25, 35, and 45 out of total 50 steps.

## 5 Conclusion

In this work, we presented VORTA, an efficient and generalized framework for accelerating diffusion transformers in video generation. VORTA reduces attention overhead by dynamically identifying attention patterns and routing them through appropriate sparse attention mechanisms. To further enhance efficiency, we introduced a bucketed coreset selection (BCS) strategy that improves the modeling of long-range dependencies. VORTA achieves a $1.76\times$ end-to-end speedup without compromising generation quality. Moreover, its high compatibility with existing acceleration techniques enables a combined speedup of up to $14.41\times$. We believe VORTA offers a practical and extensible solution, paving the way for broader adoption and future research in video generation.

**Limitations.** VORTA targets the attention mechanism, which accounts for over 75% of the total computation in high-resolution video generation. However, for tasks with short sequence lengths, such as image or low-resolution video generation, the attention overhead becomes less dominant. As a result, the potential acceleration achievable is limited. Additionally, this work focuses on the bidirectional generation paradigm. Other paradigms, such as autoregressive generation, are not directly supported and may require substantial adaptation. Additional discussion on failure cases, limitations, and border impacts is provided in Appendix C.

## Acknowledgments and Disclosure of Funding

This project is supported by the National Research Foundation, Singapore, under its NRF Professorship Award No. NRF-P2024-001.

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

# VORTA: Efficient Video Diffusion via Routing Sparse Attention

## Appendix

## Table of Contents

## A    Methodology and implementation details

### A.1    Methodological details

**Sliding tile attention.**    Section 3.2 introduced the sliding tile attention [55], the sparse attention pattern we used to model local interactions. To keep the main text concise, we explained only the 1D case. Here, we present the 3D version of the sliding tile attention mask used in our implementation, as detailed in Algorithm 1.

**Bucketed coreset selection (BCS).**    Section 3.2 also introduced the bucketed coreset selection (BCS) method, designed to reduce the computational overhead of modeling long-range interactions. In the naive setting, computing pairwise similarities across an input sequence of length $L$ incurs a quadratic complexity of $\mathcal{O}(L^2)$. Bolya and Hoffman [4], Sun et al. [40] select a subset of tokens (*e.g.*, 25%) as anchors and computing similarities between these anchors and the remaining tokens. This reduces the number of comparisons but still results in $\mathcal{O}((3L/4) \cdot (L/4)) = \mathcal{O}(L^2)$ complexity.

In contrast, BCS achieves linear complexity $\mathcal{O}(L)$ by employing a bucketing strategy. Each bucket, of size $(t, h, w)$, computes similarities between a central token and the remaining $(thw - 1)$ tokens, yielding a per-bucket cost of $\mathcal{O}(thw)$. With $L/(thw)$ such buckets, the total cost becomes $\mathcal{O}((L/(thw)) \cdot thw) = \mathcal{O}(L)$. No inter-bucket comparisons are performed, and the emperical results in Section 4.1 show that this approach is effective enough in selecting a representative subset of tokens for long-range interactions. Compared to global pairwise methods [4, 40], which require $\mathcal{O}(L^2)$ operations, BCS offers a substantially more efficient $\mathcal{O}(L)$ alternative.

### A.2    Implementation details

**Implementation details for our VORTA.**    Besides the implementation introduced in Section 4, we also provide the implementation details for our VORTA model.

For router optimization, we train on the Mixkit dataset [20] for 100 steps using a learning rate of $10^{-2}$ and a batch size of 4. Training completes in approximately one day using two H100 GPUs.

**Algorithm 1** 3D sliding tiled attention mask

---

**Input:** video size $\mathbf{v} = (F, H, W)$, tile size $\mathbf{t} = (t_F, t_H, t_W)$, window size $\mathbf{w} = (w_F, w_H, w_W)$.
 1: *// Total sequence length for self-attention*
 2: $L \leftarrow F \times H \times W$
 3: $\mathbf{M} \leftarrow \mathbf{0}_{L \times L}$
 4: *// The attention mask between the $i$-th query and the $j$-th key*
 5: **for** $i \leftarrow 1$ to $L$ **do**
 6:   **for** $j \leftarrow 1$ to $L$ **do**
 7:     *// Get tile id from token id*
 8:     $q_F, q_H, q_W \leftarrow \text{get\_tile\_id\_3d}(i, \mathbf{v}, \mathbf{t})$
 9:     $k_F, k_H, k_W \leftarrow \text{get\_tile\_id\_3d}(j, \mathbf{v}, \mathbf{t})$
10:     *// All queries within the same window share the same tile id as the window center tile id*
11:     $q_F, q_H, q_W \leftarrow \text{get\_window\_center\_id}(q_F, q_H, q_W, \mathbf{w})$
12:     *// **true** if key tile is within the local window of query tile*
13:     $m \leftarrow \text{bool}(\text{abs}(q_F - k_F) \leq w_F/2)$
14:     $m \leftarrow m$ **and** $\text{bool}(\text{abs}(q_H - k_H) \leq w_H/2)$
15:     $m \leftarrow m$ **and** $\text{bool}(\text{abs}(q_W - k_W) \leq w_W/2)$
16:     *// Save the mask*
17:     $\mathbf{M}[i, j] \leftarrow m$
18:   **end for**
19: **end for**
**Output:** sliding tile attention mask $\mathbf{M}$.

---

The sliding attention branch employs a window size of $(18, 27, 24)$, while the coreset attention branch uses a bucket size of $(2, 3, 2)$ with a coreset ratio of $r_{\text{core}} = 0.5$.

Regarding video configurations during both training and inference, HunyuanVideo [16] utilizes videos with 117 frames at a resolution of $720 \times 1280$ (720p), while Wan 2.1 [43] uses 77-frame videos at the same resolution. For rendering, HunyuanVideo outputs videos at 24 frames per second (FPS), whereas Wan 2.1 generates videos at 15 FPS, as specified in their respective repositories.

**Implementation details for baseline methods.** To evaluate efficiency, the PAB [60] is tested on 720p videos from HunyuanVideo [16], aligning with the setup used in other methods. However, processing at this resolution with PAB requires over 80 GB of GPU memory. To address this limitation, we apply sequential CPU offloading [42]. For a fair comparison, we exclude the latency caused by model loading and offloading from the reported results. Despite this, the end-to-end runtime with CPU offloading exceeds 2000 seconds per video, which is substantially slower than the original pretrained model and offers no practical efficiency advantage. The original STA kernel supports video generation only at a fixed resolution of $768 \times 1280$, which exceeds the 720p resolution used by the pretrained model. To ensure consistency in evaluation, we reimplemented the kernel using FlexAttention to support 720p video generation.

**Evaluation metrics.** To evaluate how the generated outputs differ from those of pretrained models, we use LPIPS [57] as a reference metric. However, since the pretrained models do not represent the ground truth, divergence from them does not necessarily indicate degraded performance. Multiple generated outputs may be equally valid, provided they align with the input prompts. LPIPS is only used as a comparative reference. The primary evaluation of video quality and prompt alignment should be based on VBench [11] scores.

**The improved performance of acceleration methods.** Interestingly, despite using less computation, VORTA slightly outperforms the original pretrained models in terms of VBench score. Similar findings have been reported in other acceleration methods [17, 48]. A possible explanation lies in the redundancies present in overparameterized models, which can introduce marginal negative effects. By pruning these redundancies, these acceleration methods may contribute to slight performance gains, although they are not intended for this purpose.

# B    Additional experimental results and findings

## B.1    Hyperparameters analysis

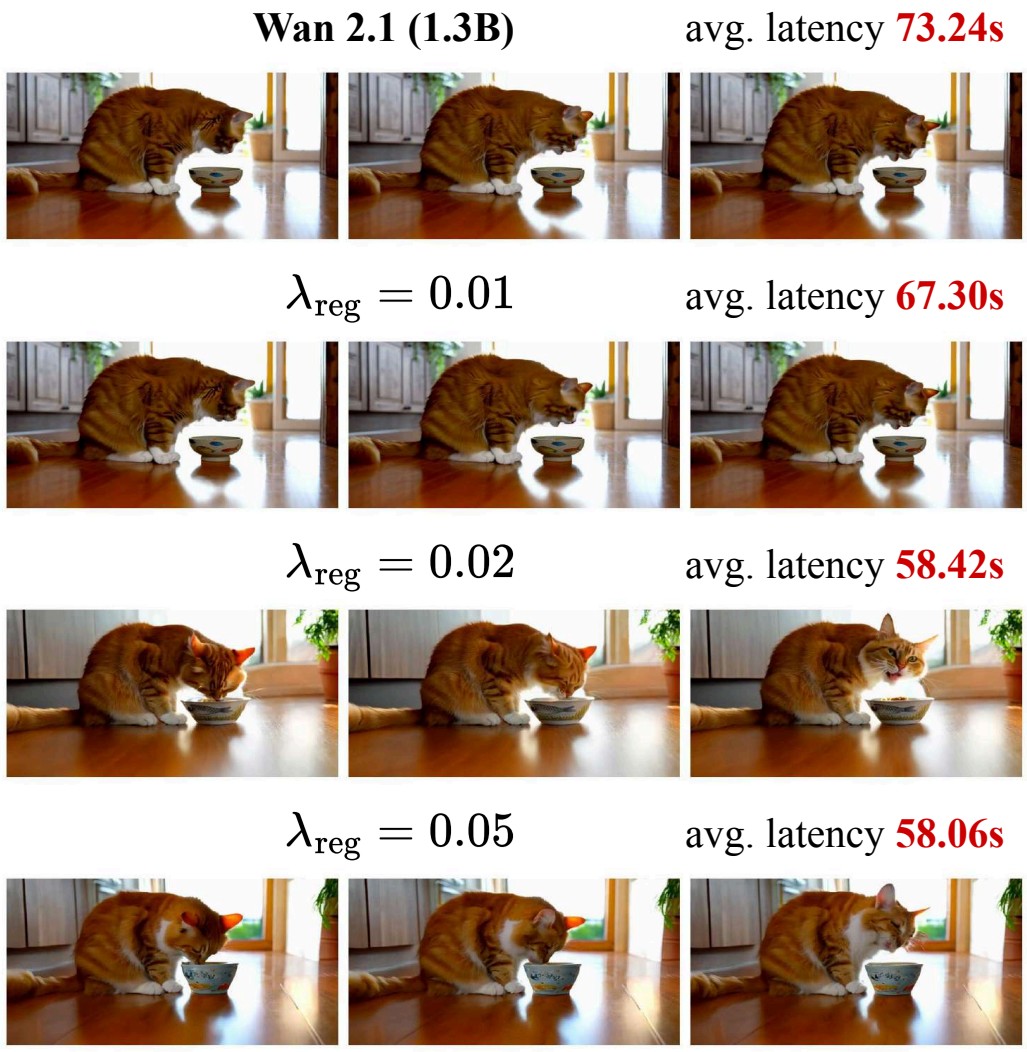

**prompt**: *a fluffy orange tabby cat ... eating from a ceramic bowl decorated with fish patterns ...*

Figure 10: Qualitative evaluation of varying the regularization weight $\lambda_{\text{reg}}$.

Figure 10 shows the effect of varying the regularization weight $\lambda_{\text{reg}}$. When $\lambda_{\text{reg}}$ is small ($\lambda_{\text{reg}} = 0.01$), the speedup is limited, and in most scenarios, the router tends to select full attention. In contrast, with a large $\lambda_{\text{reg}} = 0.05$, the video exhibits noticeable distortion (*e.g.*, the cat's head in the final frame). A moderate value of $\lambda_{\text{reg}} = 0.02$ achieves a good trade-off between acceleration and output quality.

Figure 11 provides further analysis of alternative pooling strategies in the coreset attention branch. Using average pooling with a $r_{\text{core}} = 50\%$ coreset ratio significantly underperforms compared to BCS pooling, primarily because it lacks a selection mechanism. As the coreset ratio $r_{\text{core}}$ increases, the VBench score improves; however, a ratio of $r_{\text{core}} = 50\%$ is sufficient to achieve strong performance. Higher ratios yield marginal performance gains while introducing additional computational overhead during generation.

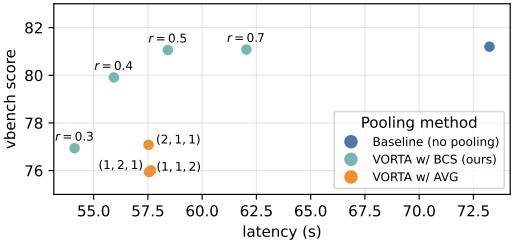

Figure 11: Effect of varying the coreset size in BCS and the kernel size in average pooling.

## B.2 Runtime analysis

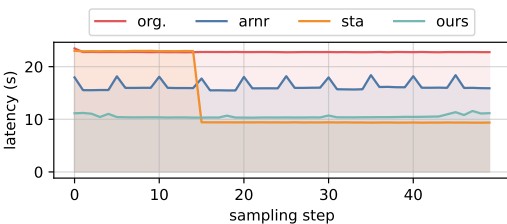

Figure 12: Per-step sampling latency on HunyuanVideo [16]. ARnR [40] yields smaller speedups overall. STA [55] shows no speedup in early timesteps.

Section 4.1 analyzed the average runtime (over diffusion steps) of each component, as shown in Figure 8. Figure 12 further presents the runtime breakdown per diffusion step. Overall, ARnR exhibits relatively high runtime, primarily due to limited acceleration in attention operations. Additionally, its periodic global similarity computation, with $\mathcal{O}(L^2)$ complexity, introduces a bottleneck. This is evident from latency spikes every five steps. In contrast, STA shows significantly higher runtime during the initial 15 steps due to the lack of early-stage acceleration, leading to reduced overall efficiency. Our VORTA achieves near-constant runtime across all steps, demonstrating stable and efficient performance.

## B.3 Attention pattern analysis

Figures 13 and 14 show the router gate values for Hunyuan [16] and Wan 2.1 (14B) [43], respectively, as supplementary results for Section 3.1.

## B.4 Qualitative comparison

In addition to the qualitative results in Figures 1 and 7 and the quantitative results in Table 1, we present further visualizations for Wan 2.1 (14B) in Figure 15. VORTA inherits the strong performance of the pretrained ViTs while offering a significant speedup.

## B.5 VBench dimensional scores

VBench evaluates the generated videos across 16 dimensions. Due to space constraints, we report only three aggregated scores in Table 1, with the complete set of scores provided in Table 4 for completeness.

## B.6 Comparison with SVG

We evaluate Spare Video Gen (SVG) [46] on a B200 GPU and compare it with our VORTA, as shown in Table 5. The comparison covers four VBench dimensions for quality and latency, as well as peak memory usage for efficiency. Since both SVG and VORTA focus on attention sparsity, minor non-attention optimizations (*e.g.* operator fusion and quantization) are disabled in SVG to ensure a

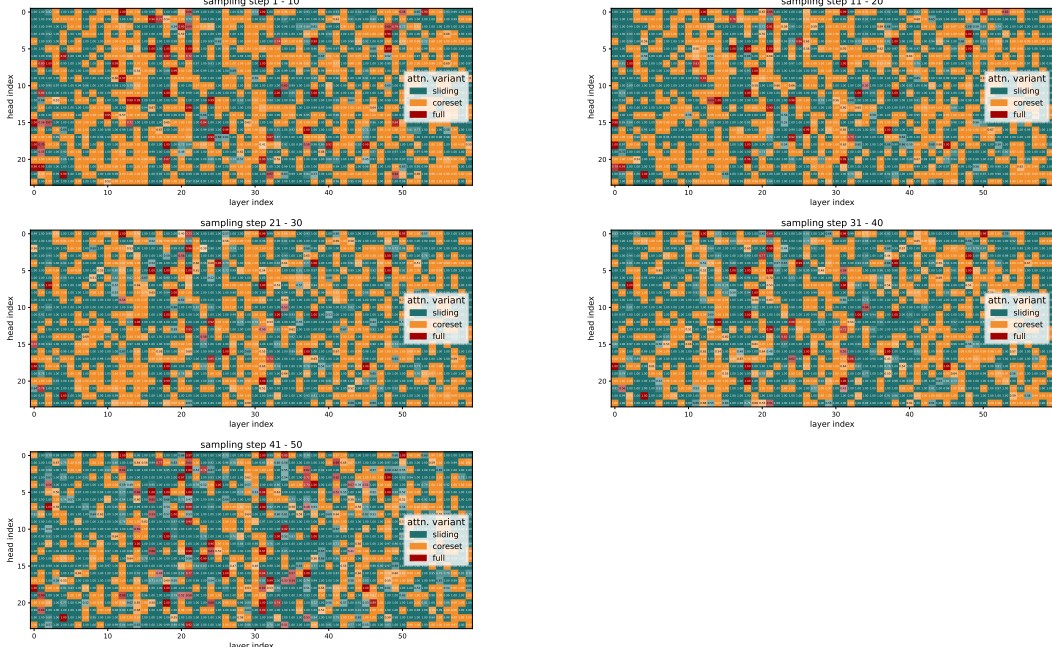

Figure 13: Optimized gate values for HunyuanVideo [16]. The 50 sampling steps are grouped into 5 equal intervals, and the averaged gate value within each interval is reported. Each color corresponds to a distinct attention branch, with color intensity indicating confidence.

fair comparison. We define attention sparsity as the ratio of skipped query-key token multiplications to the total number of query-key multiplications in standard dense attention. A higher sparsity indicates better theoretical computational efficiency.

For both Hunyuan and Wan 2.1 models, the VBench scores are statistically indistinguishable, indicating no significant differences in quality. However, VORTA consistently outperforms SVG in latency and sparsity metrics. Additionally, SVG incurs substantially higher memory usage due to its reliance on online profiling, which limits its applicability in resource-constrained environments.

## C  Limitations and border impact

### C.1  Failure cases

VORTA does not modify the pretrained parameters of VDiTs, and therefore its performance inherently depends on the quality of the underlying pretrained model. As illustrated in Figure 16, when the pretrained VDiTs fail to generate high-quality videos, resulting in distortions or non-physical outputs, VORTA exhibits similar deficiencies. In some cases, it even produces outputs of lower quality than the original VDiTs. This degradation is attributed to computations on erroneous generations, which amplify distortions in the resulting videos.

### C.2  Border impact

Generative models pose risks of producing biased, privacy-invasive, or harmful content. While our method accelerates video generation, it may also propagate such risks. It is imperative that researchers, developers, and platform providers actively assess and mitigate these potential harms to promote responsible use.

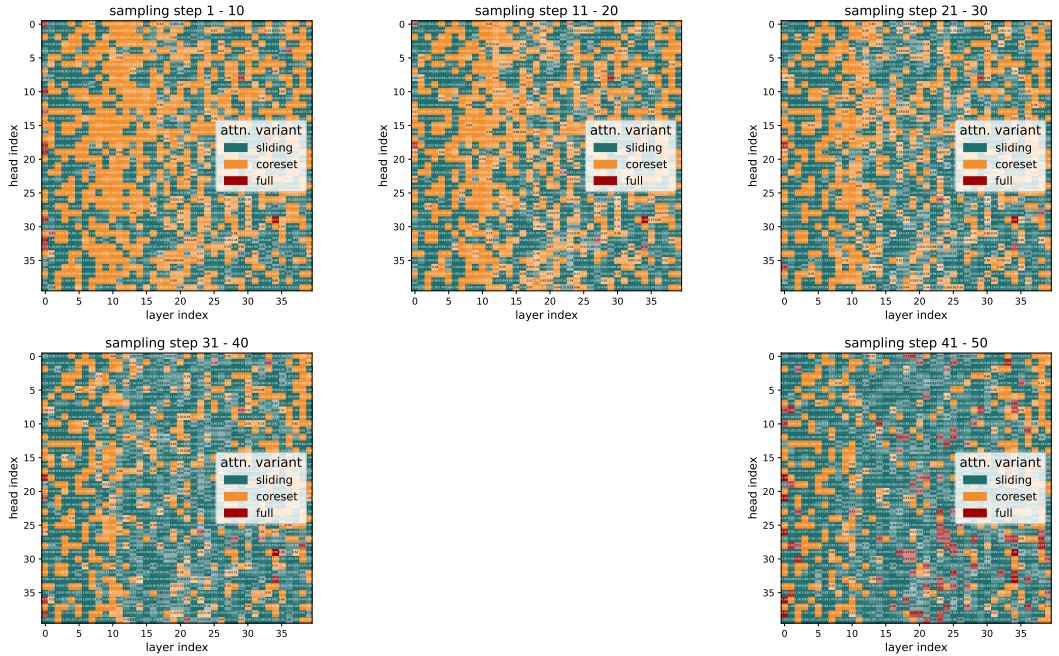

Figure 14: Optimized gate values for Wan 2.1 (14B) [43].

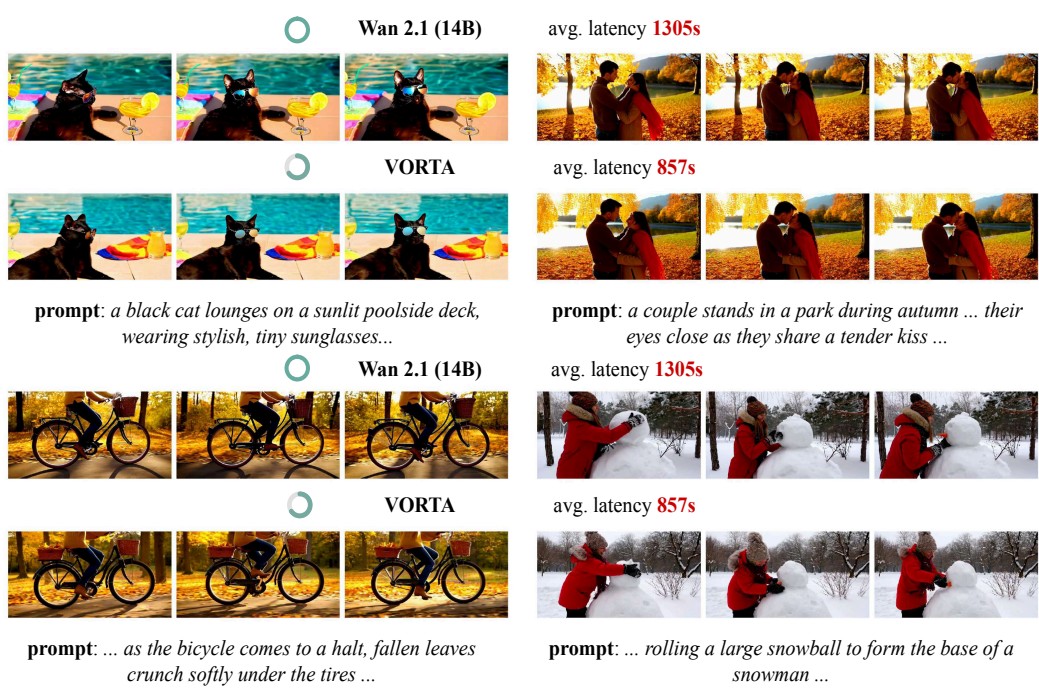

Figure 15: Qualitative comparison on Wan 2.1 (14B) [43].

Table 4: Quantitative results for VBench dimensions [11] for HunyuanVideo [16] and Wan (14B) [43].

| Method | Aesthetic Quality ↑ | Appearance Style ↑ | Background Consistency ↑ | Color ↑ | Dynamic Degree ↑ | Human Action ↑ | Imaging Quality ↑ | Motion Smoothness ↑ |
|---|---|---|---|---|---|---|---|---|
| HunyuanVideo | 62.05 | 77.88 | 93.80 | 91.35 | 38.89 | 96.00 | 63.33 | 96.98 |
| + ARnR | 62.58 | 80.36 | 93.99 | 87.54 | 39.77 | 93.73 | 63.21 | 96.12 |
| + STA | 63.87 | 79.48 | 94.47 | 86.39 | 39.58 | 97.00 | 61.92 | 96.79 |
| + PAB | 63.53 | 75.85 | 94.82 | 87.92 | 36.11 | 98.00 | 63.40 | 97.34 |
| + VORTA | 62.33 | 80.63 | 94.62 | 92.13 | 37.50 | 97.00 | 63.16 | 95.84 |
| + VORTA & PAB | 63.16 | 80.43 | 95.22 | 90.92 | 36.31 | 97.50 | 62.05 | 96.18 |
| + PCD | 62.69 | 76.21 | 94.44 | 85.94 | 31.94 | 94.00 | 63.08 | 96.46 |
| + VORTA & PCD | 61.43 | 76.92 | 94.87 | 89.33 | 35.42 | 98.00 | 62.42 | 95.11 |
| Wan 2.1 (14B) | 63.33 | 82.12 | 94.55 | 83.26 | 37.50 | 99.00 | 63.99 | 91.60 |
| + VORTA | 63.79 | 83.32 | 95.28 | 87.04 | 39.58 | 100.00 | 63.07 | 92.17 |

| Method | Mutliple Objects ↑ | Objects Class ↑ | Overall Consistency ↑ | Scene ↑ | Spatial Relationship ↑ | Subject Consistency ↑ | Temporal Flickering ↑ | Temporal Style ↑ |
|---|---|---|---|---|---|---|---|---|
| HunyuanVideo | 52.21 | 84.41 | 74.30 | 65.59 | 78.06 | 90.30 | 98.57 | 69.61 |
| + ARnR | 52.09 | 87.49 | 69.88 | 69.21 | 80.89 | 90.55 | 99.28 | 70.70 |
| + STA | 56.55 | 87.82 | 75.01 | 64.08 | 80.60 | 88.12 | 98.40 | 69.59 |
| + PAB | 56.86 | 84.97 | 74.97 | 63.91 | 78.34 | 90.89 | 98.61 | 70.50 |
| + VORTA | 58.00 | 89.56 | 74.62 | 61.87 | 78.30 | 92.43 | 98.47 | 69.44 |
| + VORTA & PAB | 59.17 | 89.90 | 75.82 | 62.99 | 77.43 | 92.27 | 97.96 | 70.29 |
| + PCD | 54.04 | 81.88 | 75.07 | 66.03 | 74.48 | 90.64 | 97.37 | 70.52 |
| + VORTA & PCD | 51.07 | 85.05 | 74.93 | 67.09 | 73.70 | 91.34 | 97.50 | 70.69 |
| Wan 2.1 (14B) | 74.62 | 86.47 | 75.49 | 64.70 | 79.16 | 91.21 | 97.69 | 71.54 |
| + VORTA | 75.76 | 88.45 | 75.09 | 63.29 | 80.28 | 90.76 | 97.78 | 70.70 |

Table 5: Quantitative comparison with SVG [46].

| Models | Sparsity (%) | VBench | | | | Latency (s) | Mem. (GB) |
|---|---|---|---|---|---|---|---|
| | | Subject ↑ | Consistency ↑ | Aesthetic ↑ | Imaging ↑ | | |
| HunyuanVideo [16] | 0.00 | 90.30 | 74.30 | 62.05 | 63.33 | 1224.22 | 47.64 |
| + SVG [46] | 80.00 | 90.59 | **75.11** | **62.73** | 63.17 | 664.48 | 71.38 |
| + VORTA | **82.65** | **92.43** | 74.62 | 62.33 | 63.16 | **568.74** | **51.15** |
| Wan 2.1 (14B) [43] | 0.00 | 91.21 | 75.49 | 63.33 | 63.99 | 1359.76 | 41.77 |
| + SVG [46] | 75.00 | 90.42 | 74.96 | 63.53 | **64.45** | 921.50 | 65.01 |
| + VORTA | **81.68** | **90.76** | **75.09** | **63.79** | 63.07 | **791.02** | **43.97** |

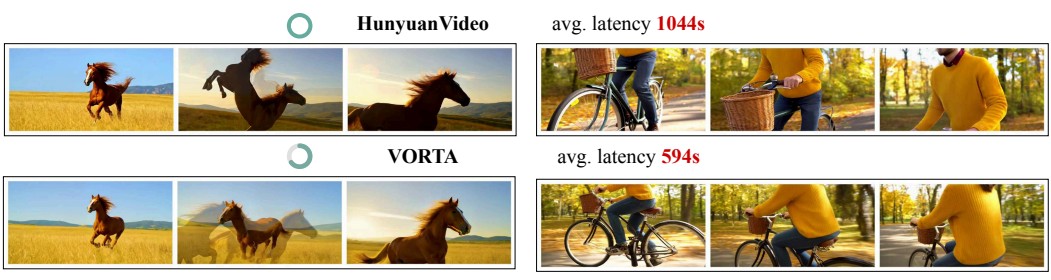

| HunyuanVideo | avg. latency **1044s** |
| VORTA | avg. latency **594s** |

**prompt**: *a horse galloping across an open field...*

**prompt**: *... as the bicycle comes to a halt, fallen leaves crunch softly under the tires ...*

**issue**: *unexpected fragment segmentation and distortion*

**issue**: *violation of physical laws*

Figure 16: Failure cases. When the pretrained model exhibits distortions or non-physical phenomena, VORTA inherits these issues. We refer the reader to the supplementary video for a more illustrative example.

