# OpenReview forum: "VORTA: Efficient Video Diffusion via Routing Sparse Attention"
_NeurIPS.cc/2025/Conference — NeurIPS 2025 poster_

### Official Review · Reviewer_Pmtr · 2025-06-29

**Clarity:** 3
**Significance:** 3
**Originality:** 3
**Rating:** 5
**Confidence:** 4

**Summary:**

VORTA is a plug-in accelerator for Video Diffusion Transformers that adds a tiny router and two sparse attention options to every self-attention block. At each diffusion step, the router, conditioned on timestep and head features, sends a head either to a sliding-tile local window or to a global branch that keeps only a coreset of representative tokens chosen inside space-time buckets, falling back to full attention only when needed. This adaptive routing turns quadratic attention into near-linear cost without retraining the generator. Tested in 720p HunyuanVideo and Wan2.1, VORTA shortens end-to-end inference time and lowers memory usage while matching or slightly improving baseline models.

**Questions:**

- The router is distilled only to replicate dense attention, yet Table 1 shows a 0.3 -0.4 VBench gain over the baseline. Please analyse the source of this improvement. Showing that the gain is statistically significant and not incidental would raise my rating.
- VBench is multi-dimensional (scoring frame fidelity, motion, text alignment, etc.) but all dimensions are produced by learned evaluators. A second, independent lens (e.g., a brief human-preference study or a metric like FVD) would strengthen the "no-quality-loss" claim.
- Add a short related-work paragraph on conditional computation, e.g. ConvNet-AIG (Veit et al., 2017) and Information Gain Trellis (Bicici et al., 2024), to acknowledge prior adaptive-routing ideas and highlight what is new (per-head, per-timestep gating for diffusion).

**Ethical Concerns:**

["NO or VERY MINOR ethics concerns only"]

**Final Justification:**

After reviewing the authors' detailed rebuttal, I am increasing my score from a 4 to a 5. The authors have exceeded expectations in addressing my concerns. They provided new experimental evidence (a human study, statistical significance tests) and insightful analysis (attention weight distribution) that have substantially strengthened the paper. My initial reservations about quality evaluation, robustness, and the source of performance gains have been fully resolved. The paper is a solid contribution to the field, and I now recommend acceptance.

**Limitations:**

yes

**Paper Formatting Concerns:**

No major formatting issues.

**Quality:**

3

**Strengths And Weaknesses:**

Strengths:
- The paper evaluates VORTA on two demanding 720p text-to-video diffusion transformers and reports end-to end latency, GPU memory, VBench sub-scores, and LPIPS, giving a well-rouded view of speed-quality trade-offs.
- Experiments show that adaptive routing alone yields about 1.7 times speed-ups without sacrificing VBench quality, and that VORTA composes additively with cached-KV and step-distillation to reach up to 14 times overall acceleration.
- Ablation studies isolate every design decision: removing either sparse branch, disabling timestep conditioning, replacing bucketed coreset selection with naive pooling, and omitting full-attention fallback. Each variant's impact on speed is quantified.
- Analysis of attention-score statistics across diffusion steps reveals that global context dominates early iterations while local detail dominates later ones; this empirical insight motivates the router and ilustrates a general principle that may guide future diffusion designs.
- Implementation details are clearly documented, making reproductions easier.

Weaknesses:
- All quality evidence comes from a single, learned-metric suite (VBench). Adding a second, independent check, either a brief human-preference study or a public metric like FVD, would make the ‘no quality loss’ claim more robust.
- The learned router, though lightweight, requires an extra distillation pass and a small video dataset. It would be good addition to the paper if they examine router robustness on out-of-distribution prompts or much longer clips.
- Memory consumption rises by around 7% or more versus the dense baseline, and the added branches increase implementation complexity, which could hinder deployment on single-GPU or resource-constrained environments.
- Related-work discussion omits earlier conditional-computation and information-gain routing methods (e.g., ConvNet-AIG, Information Gain Trellis), understating the lineage of adaptive gating ideas and making the novelty of the router appear greater than it is.

---

> ### Author Rebuttal · Authors · 2025-07-30
>
> We thank Reviewer **Pmtr** for acknowledging the clarity of our experiments, analysis, and documentation, and for providing thoughtful and constructive feedback. Your comments have been invaluable in refining our work.
>
> ---
> > **Q1.** Table 1 shows a 0.3-0.4 VBench gain over the baseline. Please analyse the source of it.
>
> **Source of Gains on the Aggregated VBench Score.** As shown in Table 3 of the Appendix, the improvement mainly stems from the "object" and "multiple objects" dimensions. Each generated video is assigned a score for each dimension, measuring the proportion of prompt-specified objects successfully detected in the video. This score reflects the semantic accuracy of the generation. Appendix Table 3 shows the averaged scores across all generated videos.
>
> **Statistical significance test of the gains.** We performed the Wilcoxon signed-rank test (a non-parametric alternative to the t-test for paired samples) on the Hunyuan scores $s_{0i}$ and the VORTA scores $s_{1i}$.
>
> - **Null hypothesis** $H_0$: The distribution of $\{s\_{0i} - s\_{1i}\}$ is symmetric about $\mu = 0$.
> - **One-sided alternative hypothesis** $H_1$: The distribution of $s_{0i} - s_{1i}$ is symmetric about $\mu < 0$.
>
> For the "object" and "multiple objects" dimensions, the p-values are 0.003 and 0.03, respectively. Both results allow us to reject $H_0$ at the 5% significance level, indicating statistically significant improvements.
>
> **Further investigation.** Hunyuan Video adopts the MMDiT [5] architecture, where text and vision tokens are concatenated and jointly processed through self-attention. Unlike prior vision diffusion models (e.g., Stable Diffusion) that commonly employ cross-attention for text conditioning, MMDiT relies solely on self-attention. So we analyze how attention weights are distributed across modalities in both dense and core-set attention. On average, the attention weights from vision queries to text keys reached 33% in core-set attention, compared to only 18% in dense attention. This suggests that the core-set selection step reduces visual redundancy and allocates more attention to semantic (textual) cues, thereby enhancing the model’s performance on semantically driven metrics. This gain does not lead to a drop in quality, as fine-grained vision-to-vision attention is less critical in components that employ core-set attention.
>
> Besides the distillation loss, which emphasizes replicating the dense attention outputs, we also employ a conditional flow-matching loss $\mathcal{L}\_\text{CFM}$ to guide learning toward the ground truth (main manuscript Equation 1). The use of core-set selection can be interpreted as a form of unstructured pruning, contributing to the observed performance.
>
> ---
> > **Q2, W1.** A second, independent lens (e.g., a brief human-preference study or a metric like FVD) would strengthen the no-quality-loss claim.
>
> FVD is known to exhibit instability and weak correlation with human evaluations [1, 2, 3, 4].
>
> So we conducted a human preference study on the generated videos. Specifically, we evaluated 100 sampled videos, where three human evaluators were asked to select the better video, or indicate a tie, between the dense model and VORTA outputs. Results show that 94% of the videos were judged as ties. These results will be incorporated in the next revision.
>
> | Method | Votes |
> | --- | --- |
> | Hunyuan Dense  | 11 (3.7%) |
> | Tie | 282 (94%) |
> | VORTA | 7 (2.3%) |
> ---
> > **W2.** It would be good addition to examine router robustness on out-of-distribution prompts or much longer clips.
>
> **VBench is out-of-distribution evaluation with respect to the training data.** The training dataset, Mixkit, consists exclusively of real-world videos. In contrast, VBench includes prompts for stylized or synthetic content (for example, beach scenes rendered in Van Gogh, oil painting, or Ukiyo-e styles). The model’s stable performance on VBench suggests robustness to out-of-distribution prompts.
>
> **Statistical test of the OOD claim.** We perform a Maximum Mean Discrepancy two-sample test using 400 CLIP embeddings from the training prompts and 400 CLIP embeddings sampled from VBench prompts, testing the null hypothesis ($H_0$) that both samples are drawn from the same distribution. The resulting p-value is 0.0009, which allows us to reject $H_0$ at the 0.1% significance level. This confirms that the VBench prompts represent a distribution shift from the training data, and the performance on VBench prompts highlights the strong generalization capability of our VORTA.
>
> VORTA can be extended to longer video clips, and additional experiments are planned. However, we observed a notable degradation in the performance of the dense model as the number of frames increases. In particular, temporal distortion becomes significant when the frame count exceeds 157. Consequently, we do not exhibit further experiments on longer clips at present.
>
> ---
> > **W3.** Memory consumption rises by around 7% or more vs the dense baseline, which could hinder deployment on single-GPU or resource-constrained environments.
>
> We observe that VORTA reaches a peak memory usage of 51 GB for 5-second 720p videos, which exceeds the capacity of many consumer GPUs. However, this is primarily due to the dense HunyuanVideo model itself, which consumes approximately 47 GB and is unsuitable for resource-constrained devices.
>
> Moreover, **VORTA requests the lowest memory among efficient attention methods** (e.g., ARnR and STA). Caching-based approaches like PAB require over 80 GB of memory, even surpassing the capacity on high-end GPUs like the H100.
>
> ---
> > **Q3, W4.** Add a short related-work paragraph on conditional computation, e.g. ConvNet-AIG and Information Gain Trellis, to acknowledge prior adaptive-routing ideas and highlight what is new.
>
> Thank you for your suggestion. We will incorporate these references into the related work section to better situate our contributions within the existing literature and enhance the overall coherence of the paper.
>
> ---
> **Reference**
>
> 1. Singer, et al. “Make-A-Video: Text-to-Video Generation without Text-Video Data.” ICLR (2023)
> 2. Wang, et al. “LAVIE: High-Quality Video Generation with Cascaded Latent Diffusion Models.” IJCV (2024)
> 3. Polyak, et al. “Movie Gen: A Cast of Media Foundation Models”. preprint arXiv: 2410.13720 (2024)
> 4. Ge, et al. “On the Content Bias in Fréchet Video Distance.” CVPR (2024)
> 5. Esser, et al. "Scaling Rectified Flow Transformers for High-Resolution Image Synthesis." ICML (2024)

---

### Official Review · Reviewer_BsHH · 2025-06-29

**Clarity:** 3
**Significance:** 2
**Originality:** 2
**Rating:** 4
**Confidence:** 4

**Summary:**

This paper proposes VORTA, a sparse attention technique with a router to speed up the inference stage of Video Diffusion Models. This paper proposes three attention variations: full attention, sliding window attention, and coreset attention for each head. They further propose a router to classify each attention head into one of these attention patterns, and apply distillation loss and regularization loss to punish the excessive usage of full attention. VORTA exhibits strong quality and speedup performance on HunyuanVideo and Wan 2.1, demonstrating its effectiveness.

**Questions:**

1. What's the average attention sparsity during the generation process for sliding window attention and coreset attention?
2. What is the value of r_core for BCS module? How does this value affect the final performance?
3. What will the attention maps look like for the 0.2% full attention heads? More visualization results for these attention heads will help to understand the model's behavior.

**Ethical Concerns:**

["NO or VERY MINOR ethics concerns only"]

**Final Justification:**

The authors' rebuttal is clear and solid. They have addressed my concerns, in terms of explanation, experimental validation, and numerical calculation. Therefore I raised my score.

**Limitations:**

Yes

**Quality:**

2

**Strengths And Weaknesses:**

Strengths:
1. The design of the attention router, along with the introduction of the distillation and regularization loss term, is innovative and contributes to the novelty of the approach.
2. The proposed method is flexible and can be effectively integrated with other techniques, such as PAB, to achieve additional speedup.

Weaknesses:
1. The coreset attention pattern requires further empirical validation. It remains unclear how performance would be affected if alternative attention patterns—such as the temporal attention used in Sparse VideoGen[1]—were employed.
2. The paper does not provide comparisons against other baselines using the Wan 2.1 model, which limits the assessment of its relative performance.
3. Refer to the questions section.

[1] Xi, Haocheng, et al. "Sparse VideoGen: Accelerating Video Diffusion Transformers with Spatial-Temporal Sparsity." arXiv preprint arXiv:2502.01776 (2025).

---

> ### Author Rebuttal · Authors · 2025-07-30
>
> We thank the Reviewer **BsHH** for acknowledging the motivation, novelty, and extensibility of our approach. Below, we provide responses to the questions and concerns raised. For clarity, we have reordered the responses to improve readability and interpretation.
>
> ---
>
> > **Q2.** What is the value of $r_\mathrm{core}$ for BCS module? How does this value affect the final performance?
>
> **Explanation of $r_\mathrm{core}$.** The core-set ratio $r_\mathrm{core}$ denotes the proportion of tokens retained for attention computation after applying BCS, relative to the full sequence length (main manuscript Section 3.2). For example, given a sequence of 100 tokens, setting $r_\mathrm{core} = 0.4$ means that 60 tokens are deduplicated during BCS, and only 40 tokens participate in the subsequent attention computation.
>
> **Impact of varying** **$r_\mathrm{core}$.** Reducing $r_\mathrm{core}$ leads to more aggressive compression and faster inference, at the potential cost of output quality. When $r_\mathrm{core} = 1$, no tokens are removed, and the computation is equivalent to standard dense attention. As $r_\mathrm{core}$ decreases to 0.5, video quality remains nearly unchanged, as indicated by VBench scores and human case study. However, further reductions below 0.5 result in noticeable quality degradation, despite latency improvements. Detailed results are provided in Appendix B.1 and Figure 10 of the main manuscript.
>
> As a consequence, we use $r_\mathrm{core} =0.5$ in all experiments (main manuscript Appendix A.2). We will define this variable more explicitly in the next revision.
>
> ---
>
> > **Q1.** What's the average attention sparsity during the generation process for sliding window attention and core-set attention?
>
> **Definition of sparsity.** We define attention sparsity as the ratio of skipped query-key token multiplications to the total number of query-key multiplications in standard dense attention. A higher sparsity indicates better theoretical computational efficiency.
>
> **HunyuanVideo’s overall sparsity is 82.65%.** HunyuanVideo generates video tokens with shape (30, 45, 80)
>
> - For sliding window attention, we use a window size of (18, 27, 24). The resulting sparsity is computed as
>     - $$1 - \frac{(18 \times 27 \times 24) \times (30 \times 45 \times 80)}{(30 \times 45 \times 80)^2} = 0.892$$
> - For core-set attention with $r_\mathrm{core} =0.5$. The sparsity is computed as
>     - $$1 - \frac{(0.5 \times 30 \times 45 \times 80 )^2}{(30 \times 45 \times 80)^2} = 0.75$$
> - According to the routing statistics (visualized in the main manuscript Figures 8 and 12), full attention, sliding attention, and core-set attention are used in 0.49%, 56.47%, and 43.04% of the cases, respectively. The resulting overall sparsity is **82.65%**.
>
> **Wan 2.1’s overall sparsity is 81.68%.** Wan 2.1 generates video tokens with shape (20, 45, 80)
>
> - For sliding window attention, we use a window size of (15, 27, 24). The resulting sparsity is computed as
>     - $$1 - \frac{(15 \times 27 \times 24) \times (20 \times 45 \times 80)}{(20 \times 45 \times 80)^2} = 0.865$$
> - For core-set attention with $r_\mathrm{core} =0.5$. The sparsity is computed as
>     - $$1 - \frac{(0.5 \times 20 \times 45 \times 80 )^2}{(20 \times 45 \times 80)^2} = 0.75$$
> - According to the routing statistics (visualized in main manuscript Figure 13), full attention, sliding attention, and core-set attention are used in 0.23%, 59.61%, and 40.16% of the cases, respectively. The resulting overall sparsity is **81.68%**.
>
> ---
>
> > **W2.** The paper does not provide comparisons against other baselines using the Wan 2.1 model, which limits the assessment of its relative performance.
>
> **Comparison with SVG on Hunyuan and Wan 2.1.** We evaluate Spare Video Gen (SVG) on a B200 GPU and compare it with our VORTA, as shown in Tables 1 and 2. The comparison covers four VBench dimensions for quality and latency, as well as peak memory usage for efficiency. Since both SVG and VORTA focus on attention sparsity, minor non-attention optimizations (e.g., operator fusion and quantization) are disabled in SVG to ensure a fair comparison.
>
> Table 1
>
> |  | Latency (s) ↓ | Sparsity ↓ | Memory (GB) ↓ | Subject ↑ | Consistency ↑ | Aesthetic ↑ | Imaging ↑ |
> | --- | --- | --- | --- | --- | --- | --- | --- |
> | HunyuanVideo | 1224.22 | 0% | 47.64 | 90.30 | 74.30 | 62.05 | 63.33 |
> | + SVG | 664.48 | 80% | 71.38 | 90.59 | **75.11** | **62.73** | 63.17 |
> | + VORTA | **568.74** | **82.65%** | **51.15** | **92.43** | 74.62 | 62.33 | 63.16 |
>
> Table 2
>
> |  | Latency (s) ↓ | Sparsity ↓ | Memory (GB) ↓ | Subject ↑ | Consistency ↑ | Aesthetic ↑ | Imaging ↑ |
> | --- | --- | --- | --- | --- | --- | --- | --- |
> | Wan 2.1 | 1359.76 | 0% | 41.77 | 91.21 | 75.49 | 63.33 | 63.99 |
> | + SVG | 921.50 | 75% | 65.01 | 90.42 | 74.96 | 63.53 | **64.45** |
> | + VORTA | **791.02** | **81.68%** | **43.97** | **90.76** | **75.09** | **63.79** | 63.07 |
>
> For both Hunyuan and Wan 2.1 models, the VBench scores are statistically indistinguishable, indicating no significant differences in quality. However, VORTA consistently outperforms SVG in latency and sparsity metrics. Additionally, SVG incurs substantially higher memory usage due to its reliance on online profiling, which limits its applicability in resource-constrained environments.
>
> ---
>
> > **W1.** The coreset attention pattern requires further empirical validation. It remains unclear how performance would be affected if alternative attention patterns—such as the temporal attention used in SVG—were employed.
>
> **VORTA is designed for SOTA foundational models.** Earlier video diffusion models adopted factorized spatio-temporal (2+1D) attention [1, 2, 3]. Subsequent works [4, 5, 6] transitioned to full 3D attention, demonstrating superior generation quality and establishing 3D attention as the mainstream architecture in video foundation models. Our method, VORTA, focuses on optimizing inference efficiency for these 3D attention models. We exclude older and less competitive attention mechanisms from our scope due to their diminished relevance and performance.
>
> **VORTA outperforms other acceleration methods with sparse attention pattern.** Similar to VORTA, SVG also aims to improve inference efficiency via sparse attention. However, as shown in experiment results above, VORTA achieves better efficiency without compromising performance.
>
> **VORTA can integrate with various acceleration methods in a plug-and-play manner.** Additionally, VORTA integrates well with other orthogonal efficiency techniques, such as PAB and PCD, as shown in Section 4 of the main manuscript. Incorporating another sparse attention methods would introduce significant engineering complexity with limited practical benefit.
>
> Given the constraints of the rebuttal period, we prioritize experiments and analyses with greater impact.
>
> ---
>
> > **Q3.** What will the attention maps look like for the 0.2% full attention heads? More visualization results for these attention heads will help to understand the model's behavior.
>
> We analyzed the attention maps of the full attention heads and identified several notable patterns. Due to the 2025 rebuttal policy, we cannot include figures at this stage, so we describe the key observations textually.
>
> For queries and keys within the same frame $f$, the attention pattern typically exhibits a strong diagonal, indicating that each token predominantly attends to itself. Additionally, we observe several vertical, non-uniformly spaced slashes, suggesting that certain key positions are globally important across the entire frame. Notably, the positions of these vertical lines vary with different prompts, forming a composite attention pattern resembling “| | |” + “\”.
>
> When querying tokens in frame $f$ with keys in adjacent frames $f \pm i$, a similar pattern persists. However, as $i$ increases, the overall attention weights decrease. In particular, the vertical components decay more rapidly than the diagonal ones.
>
> We will include this analysis and the corresponding figure in the next revision to provide additional insights for readers.
>
> ---
>
> **Reference**
>
> 1. Singer, et al. “Make-A-Video: Text-to-Video Generation without Text-Video Data.” ICLR (2023)
> 2. Wang, et al. “LAVIE: High-Quality Video Generation with Cascaded Latent Diffusion Models.” IJCV (2024)
> 3. Ma, et al. “Latte: Latent Diffusion Transformer for Video Generation.” TMLR (2025)
> 4. Polyak, et al. “Movie Gen: A Cast of Media Foundation Models.” preprint arXiv: 2410.13720 (2024)
> 5. Kong, et al. “HunyuanVideo: A Systematic Framework For Large Video Generative Models.” preprint arXiv: 2412.03603 (2024)
> 6. Wang, et al. “Wan: Open and Advanced Large-Scale Video Generative Models.” preprint arXiv: 2503.20314 (2025)

---

> > ### Comment · Reviewer_BsHH · 2025-08-05
> >
> > Thanks for the rebuttal; the authors have clearly addressed my concerns. I raised my score accordingly.
> >
> > I also want to suggest that the authors include PSNR, SSIM, and LPIPS metrics in their paper. In my past experiments, I found that these metrics are more representative than VBench scores.

---

> > > ### Author Response · Authors · 2025-08-06
> > >
> > > Dear reviewer BsHH,
> > >
> > > We would like to express our sincere gratitude for your constructive suggestions. We will include the additional analysis and findings in the next revision.
> > >
> > > Many thanks for the time and effort you took to review our work.
> > >
> > > The Authors

---

### Official Review · Reviewer_m2pg · 2025-06-30

**Clarity:** 2
**Significance:** 2
**Originality:** 2
**Rating:** 3
**Confidence:** 4

**Summary:**

The paper presents VORTA, an acceleration framework for Video Diffusion Transformers (VDiTs) that improves efficiency without sacrificing performance. VORTA introduces a sparse attention mechanism to capture long-range dependencies and a dynamic routing strategy to select optimal attention variants. This approach achieves a 1.76× speedup in video generation and can integrate with other methods, reaching up to a 14.41× speedup.

**Questions:**

Please refer to the weakness section.

**Ethical Concerns:**

["NO or VERY MINOR ethics concerns only"]

**Final Justification:**

The author’s response addresses part of my concerns. However, the key issue remains that some assumptions in the paper are not properly justified. Despite multiple rounds of discussion, the author’s explanations are provided without direct observations or empirical evidence, relying instead on intuitive new “assumptions” that lack clear validation. Therefore, I maintain my original stance and lean towards a weak rejection.

**Limitations:**

Yes

**Quality:**

2

**Strengths And Weaknesses:**

Strength
- Adaptability: The dynamic routing mechanism is tailored for adaptive sparse scheme for different diffusion configurations and backbones.
- Compatibility with other compression techniques: VORTA could be with other existing acceleration techniques, like caching and step distillation.
- Detailed description of implementation details in the Appendix.

Weakness
My main concern lies in the lack of clarity regarding the motivation, applicability, and effectiveness of some proposed methods.
- Bucketed Core set Selection (BCS):
  - While detailed pseudo-code is provided, it only describes how to determine the core set attention sparse mask. However, it is unclear whether this sparse mask is shared across different conditions and time steps. Additionally, it remains ambiguous whether the BCS is performed online for each attention head.
  - If BCS is performed online, the overhead analysis appears to be missing. This process involves multiple steps, which could introduce significant overhead. If the BCS sparse mask is determined offline, the paper should justify whether such a sparse mask can generalize across multiple time steps and prompts.
  - The assumption underlying the design of BCS is that the "center token is very similar to its neighboring tokens," which, as the authors state, "will not always hold." When this assumption does not hold, the claim that "the remaining tokens could complement the necessary information" needs proper justification. Furthermore, there is a lack of analysis to validate the assumption behind BCS and to assess the effectiveness of BCS itself.
- Dynamic Routing:
  - The motivation for employing training-based dynamic routing requires further justification. The necessity of the routing design should be demonstrated by showing that the optimal compression plan varies significantly across different conditions. Even in such cases, the advantage of introducing an additional training-based router, rather than simply adopting different plans or heuristic rules for different conditions, needs to be addressed.
  - Additionally, the generalization ability of the optimized router should be thoroughly verified to demonstrate its effectiveness.

---

> ### Author Rebuttal · Authors · 2025-07-30
>
> We sincerely thank the Reviewer **m2pg** for acknowledging the adaptability, extensibility, and clarity of our implementation, as well as for the insightful comments and questions. Below, we provide detailed responses to the raised concerns.
>
> ---
>
> > **W1.** It is unclear whether this sparse mask is shared across different conditions and time steps. Additionally, it remains ambiguous whether the BCS is performed online for each attention head.
>
> For the core-set attention, a sparse mask is dynamically computed for each attention head, varying across conditions and time steps. This design enables the method to adapt flexibly to diverse input patterns.
>
> ---
>
> > **W2.** If BCS is performed online, the overhead analysis appears to be missing. This process involves multiple steps, which could introduce significant overhead.
>
> We present a detailed component-wise overhead analysis in Table 1 below. Specifically, we report the latency for a single sampling step (i.e., one forward pass of the model) averaged over 10 prompts and 50 time steps. All layers and attention heads are configured to use the same attention mechanism for benchmarking. We evaluate three strategies: dense (full) attention, sliding window attention, and core-set attention.
>
> Table 1
>
> |  | Attention | Attention related | Router | BCS | Feed-Forward MLP | Others |
> | --- | --- | --- | --- | --- | --- | --- |
> | Dense | 21197.99 ($\pm 18.52$) | 2912.12 ($\pm 0.74$) | - | - | 269.91 ($\pm 0.14$) | 2.99 ($\pm 0.03$) |
> | Sliding window | 7215.30 ($\pm 14.78$) | 2913.28 ($\pm 0.69$) | 87.23 ($\pm 0.06$) | - | 268.96 ($\pm 0.13$) | 2.98 ($\pm 0.03$) |
> | Core-set | 5647.17 ($\pm 13.40$) | 2911.67 ($\pm 0.72$) | 87.24 ($\pm 0.05$) | 862.73 ($\pm 0.59$) | 267.81 ($\pm 0.11$) | 3.00 ($\pm 0.03$) |
>
> Note:
>
> - All latency values in Table 1 are in milliseconds; values in parentheses denote standard deviation.
> - “Attention-related” includes all operations associated with attention computation except the attention itself (e.g., QKV projections, output projections, and QK normalization).
> - “Others” include operations outside the transformer blocks (e.g., patchification, unpatchification, and ROPE initialization).
> - Latency measurements on HunyuanVideo were performed using a B200 GPU, which was the only available device at the time.
>
> The results indicate that, although the proposed BCS strategy and router module introduce slight additional overhead, this cost is negligible compared to the reduction in attention computation latency.
>
> An overhead analysis of the entire sampling process, conducted on HunyuanVideo with an H100 GPU, is provided in Figure 7 of the main manuscript. And a theoretical complexity analysis of the BCS is detailed in Appendix A.1 of the main manuscript.
>
> ---
>
> > **W3.** The assumption underlying the design of BCS is that the "center token is very similar to its neighboring tokens," which, as the authors state, "will not always hold." When this assumption does not hold, the claim that "the remaining tokens could complement the necessary information" needs proper justification. Furthermore, there is a lack of analysis to validate the assumption behind BCS and to assess the effectiveness of BCS itself.
>
> **Assumption and design of BCS.** In BCS, the design that we select the center token as an anchor to compute similarity scores with surrounding tokens is motivated by the assumption that “the center token is similar to its surrounding tokens”.
>
> **What if this assumption does not hold**
>
> - When fine-grained details are not critical and the model primarily captures high-level semantic features such as contours, the loss introduced by BCS has a negligible impact.
> - When fine-grained details are important, the router is trained to select alternative attention branches by replicating the output of dense attention ($\mathcal{L}\_\text{distill}$, main manuscript Equation 8) and aligning with the ground-truth denoising direction ($\mathcal{L}\_\text{CFM}$, main manuscript Equation 1).
>
> **Experimental results show the practical effectiveness of BCS.** Experiments have shown that VORTA achieves lossless video generation compared to the dense model. Notably, approximately 40% of attention heads adopt the core-set attention strategy (visualized in the main manuscript in Figures 8, 12, and 13). These findings empirically validate the BCS assumption across diverse settings and demonstrate its effectiveness for lossless acceleration.
>
> **BCS is much better than naive pooling.** BCS discards the most similar tokens and retains the least similar ones, which maintains "the necessary information". In contrast, common strategies like average pooling do not account for cases where the similarity assumption fails, often resulting in degraded performance, as shown empirically in main manuscript Section 4.3.
>
> The ablation studies comparing BCS to other pooling methods are presented in Section 4.3 and Table 2 of the main manuscript. A sensitivity analysis is further provided in Appendix B.1 and Figure 10 of the main manuscript.
>
> ---
>
> > **W4.** The motivation for employing training-based dynamic routing requires further justification. The necessity of the routing design should be demonstrated by showing that the optimal compression plan varies significantly across different conditions. Even in such cases, the advantage of introducing an additional training-based router, rather than simply adopting different plans or heuristic rules for different conditions, needs to be addressed.
> >
>
> **Our router is more flexible and adaptable than heuristic rules.** Employing heuristic rules to determine the routing strategy is of high complexity. Methods such as ARnR [1] and STA [2] rely on offline profiling and heuristics to select sparse strategies. These approaches typically require multiple sampling runs to determine the strategy for a single configuration (e.g., sampling with UniPC [3] with 50 sampling steps), which increases overhead.
>
> In practice, dense models are frequently used with varying sampling configurations to balance quality and speed. These configurations include:
>
> 1. Adjusting sampling hyperparameters (e.g., modifying the flow shift),
> 2. Changing the number of sampling steps (e.g., from 50 to 30), and
> 3. Switching the scheduler (e.g., from UniPC [3] to DPM++ [4])
>
> We aim to preserve this flexibility of dense models. Heuristic methods [1, 2] must re-profile and re-tune their strategy whenever any sampling configuration is modified. This reconfiguration process can take several hours on a single GPU due to the high computational cost of dense sampling. In contrast, our VORTA router generalizes across different sampling settings without requiring any changes or additional overhead, owing to its lightweight training paradigm.
>
> **Quantitative evaluation on diverse sampling configurations.** We conduct additional experiments on Wan, replacing the default UniPC scheduler (50 steps) with the DPM++ scheduler (30 steps). Table 2 below shows the results across four VBench metrics for quality and latency, as well as peak memory usage for efficiency.
>
> Table 2
>
> | Scheduler | Method | Latency (s) ↓ | Memory (GB) ↓ | Subject ↑ | Consistency ↑ | Aesthetic ↑ | Imaging ↑ |
> | --- | --- | --- | --- | --- | --- | --- | --- |
> | UniPC (50) | Wan 2.1 | 1359.76 | 41.77 | 91.21 | 75.49 | 63.33 | 63.99 |
> | UniPC (50) | + VORTA | 791.02 | 43.97 | 90.76 | 75.09 | 63.79 | 63.07 |
> | DPM++ (30) | Wan 2.1 | 817.00 | 41.77 | 89.72 | 74.54 | 64.43 | 63.99 |
> | DPM++ (30) | + VORTA | 440.76 | 43.97 | 91.82 | 75.13 | 64.32 | 63.95 |
>
> **Note**: *Latency measurements were performed using a B200 GPU, which was the only available device at the time.*
>
> The consistent, lossless performance of VORTA relative to the dense model confirms its ability to generalize efficiently without added cost, an advantage enabled by its lightweight training, which heuristic-based methods like STA and ARnR do not achieve.
>
> These motivations and results will be incorporated in more detail in the next revision.
>
> ---
>
> > **W5.** The generalization ability of the optimized router should be thoroughly verified to demonstrate its effectiveness.
>
> The results in W4 demonstrate that the optimized router generalizes well across different schedulers and sampling steps.
>
> **VBench is out-of-distribution evaluation with respect to the training data.** The training dataset, Mixkit, consists exclusively of real-world videos. In contrast, VBench includes stylized or synthetic content (for example, beach scenes rendered in Van Gogh, oil painting, or Ukiyo-e styles). So, evaluation on VBench (main manuscript Section 4) also confirms its generalization capability to out-of-distribution prompts.
>
> **The OOD claim is statistically significant.** To statistically assess this, we perform a Maximum Mean Discrepancy two-sample test using 400 CLIP embeddings from the training prompts and 400 CLIP embeddings sampled from VBench prompts, testing the null hypothesis ($H_0$) that both samples are drawn from the same distribution. The resulting p-value is 0.0009, which allows us to reject $H_0$ at the 0.1% significance level. This confirms that the VBench prompts represent a distribution shift from the training data, and the performance on VBench prompts highlights the strong generalization capability of our VORTA.
>
> **Reference**
>
> [1] Sun, et al. “AsymRnR: Video Diffusion Transformers Acceleration with Asymmetric Reduction and Restoration.” ICML (2025)
>
> [2] Zhang, et al. “Fast Video Generation with Sliding Tile Attention.” ICML (2025)
>
> [3] Zhao, et al. “UniPC: A Unified Predictor-Corrector Framework for Fast Sampling of Diffusion Models.” NIPS (2023)
>
> [4] Lu, et al. “DPM-Solver++: Fast Solver for Guided Sampling of Diffusion Probabilistic Models.” ICLR (2023)

---

> ### Comment · Reviewer_m2pg · 2025-08-05
>
> Thank the authors for the response, which addresses some of my concerns. However, I still have a few points that I would appreciate further clarification on:
> - As shown in the table in response to W2, the cost of coreset selection accounts for around 26% of the dense attention and nearly half of the sliding window attention. This seems significant and may not be accurately described as "slight additional overhead," especially when compared to the reduction in attention computation latency (please let me know if I’ve misunderstood this).
> - Regarding the response to W3, I believe a more thorough justification is needed. The explanation, which frames the issue in terms of two binary cases—"When fine-grained details are not critical"—is not fully convincing. Additionally, I would appreciate more clarity on how to measure when "fine-grained details are not critical." The experimental results provided seem somewhat indirect and don’t fully validate the assumptions of the BCS.

---

> > ### Author Response · Authors · 2025-08-05
> >
> > ### **P1: Clarification on Misinterpretation of Bucketed Core-set Selection (BCS) Latency**
> >
> > As shown in the table provided in response to W2 (attached below for convenience), each **row** corresponds to the computational cost of **an attention branch**, and each **column** corresponds to the cost of a **specific component**.
> >
> > |  | Attention | Attention related | Router | Bucketed Core-set Selection (BCS) | Feed-Forward MLP | Others |
> > | --- | --- | --- | --- | --- | --- | --- |
> > | Dense attention branch | 21197.99 ($\pm 18.52$) | 2912.12 ($\pm 0.74$) | - | - | 269.91 ($\pm 0.14$) | 2.99 ($\pm 0.03$) |
> > | Sliding window attention branch | 7215.30 ($\pm 14.78$) | 2913.28 ($\pm 0.69$) | 87.23 ($\pm 0.06$) | - | 268.96 ($\pm 0.13$) | 2.98 ($\pm 0.03$) |
> > | Core-set attention branch | 5647.17 ($\pm 13.40$) | 2911.67 ($\pm 0.72$) | 87.24 ($\pm 0.05$) | 862.73 ($\pm 0.59$) | 267.81 ($\pm 0.11$) | 3.00 ($\pm 0.03$) |
> >
> > The cost of core-set selection is 862.73 milliseconds per transformer forward pass, whereas the cost of dense attention is 21,197.99 milliseconds. This results in a cost ratio of approximately **4%** ($862.73 / 21,197.99$), **not 26%** as previously stated in the comment.
> >
> > For the BCS branch, the attention cost is 5,647.17 milliseconds. Therefore, the latency reduction compared to dense attention is $21,197.99 - 5,647.17 = 15,550.82$ milliseconds. The **ratio** of **attention latency saved** to the **latency introduced by BCS** is approximately **21:1** ($15,550.82 / 862.73$).
> >
> > ---
> >
> > ### **P2: Fine-Grained Details Are Less Critical in Certain Cases**
> >
> > We appreciate your suggestion. In response, we provide both intuitive explanation and quantitative results to support our claim that *"fine-grained details are less critical in certain cases"*.
> >
> > **Intuitive Explanation**
> >
> > As illustrated in Figure 3 of the main manuscript, early-stage sampling primarily captures high-level semantic structures. In such stages, details are absent, so less critical.
> >
> > This is further supported by our empirical validation in the same figure: even when we intentionally apply pooling followed by unpooling in the flow predictions, the final outputs remain largely unaffected. In other words, the loss of fine details is less critical at this stage.
> >
> > **Quantitative Validation**
> >
> > ***Definition*** In the main manuscript and previous discussion:
> >
> > - **Tokens representing fine-grained information** are those that capture highly specific details and cannot be easily substituted by others (e.g., fine textures). These tokens typically exhibit low similarity to other tokens due to their uniqueness.
> > - **Tokens representing coarse-grained information**, on the other hand, convey more general or redundant content (e.g., layout and outline). Such tokens usually show high similarity to other tokens.
> >
> > We compute intra-sequence cosine similarity among query tokens (i.e., for each token, its cosine similarity with the remaining tokens in the same sequence) across both core-set and non-core-set attention heads as determined by the router. Statistics aggregated over UniPC sampling steps and all layers of HunyuanVideo are summarized below:
> >
> > | Attention Head Type | Cosine Similarity **Mean** (**Std**) |
> > | --- | --- |
> > | Core-set attention | $0.879 (\pm 0.116)$ |
> > | Dense and sliding attention | $0.711 (\pm 0.119)$ |
> >
> > The core-set attention heads show significantly higher intra-sequence similarity, indicating a high degree of feature redundancy and less fine-grained information. Hence, these results support the claim that in such heads, fine-grained information is less critical.

---

> ### Comment · Reviewer_m2pg · 2025-08-07
>
> Thank you for the clarification. Despite comparing the overhead with the “latency improvement it brings” is not a conventional approach, the comparison with dense attention more relevant to address this concern.
>
> Regarding point P2, the author provides an intuitive explanation for “why fine-grained details are critical in certain cases,” only partially addresses my question. The major concern, however, remains with W3. The author’s explanation of “What if this assumption does not hold?” is based on purely intuitive claims and lacks empirical verification. This issue persists in several parts of the author’s response, which undermines the overall persuasiveness of their argument. For instance, the categorization of “tokens representing fine-grained details” as those “exhibiting low similarity with other tokens” is made without proper justification. The assumption are explained without direct observation or empirical evidence, but are explained through intuitive new “assumptions” without clear evidence.

---

> > ### Author Response · Authors · 2025-08-07
> >
> > Thank you for your response.
> >
> > ### **P3. The Table in W2 presents a detailed comparison between dense attention and core-set attention, including module-level breakdowns.**
> >
> > As noted in the previous comment, the latency difference between the **core-set and dense attention operation alone** is:
> >
> > **4.0% = (862.73 / 21197.99)**
> >
> > The total latency difference between the **core-set and dense branch** is:
> >
> > **40.1% = (5647.17 + 2911.67 + 87.24 + 862.73 + 267.81 + 3.00) / (21197.99 + 2912.12 + 269.91 + 2.99)**
> >
> > Similar statistics have been reported in recent related works such as STA [1] and Sparse Video Gen [2] . We are happy to provide any additional statistics if needed.
> >
> > ---
> >
> > ### **P4. The assumptions behind BCS and its practical performance**
> >
> > Thanks for your comments.
> >
> > **The assumption underlying similarity-based token selection is not merely intuitive; it is supported by logical reasoning.**
> >
> > High intra-sequence similarity suggests that certain tokens encode highly similar information, in other words these similar token convey redundant information [3, 4], which we refer to as coarse-grained information. In contrast, tokens of low intra-sequence similarity (low-redundancy) sequences encode distinct information, which we refer to as fine-grained details.
> >
> > Therefore, intra-sequence similarity serves as a practical empirical indicator for evaluating the validity of the similarity-based token selection assumption. Based on the statistical results in P2, the assumption is valid in practice.
> >
> > **The design of using center token as anchor in Bucketed Core-set Selection (BCS).**
> >
> > While similarity-based selection has proven effective in prior works, it incurs significant computational overhead due to pairwise similarity calculations. To address this, we use only the center token as an anchor to compute similarities with surrounding tokens, reducing the complexity from $O(n^2)$ to $O(n)$.
> >
> > In BCS, it is sufficient for the center to be similar to the top $(1−r_\text{core})$ fraction of the most similar tokens. If these top $(1−r_\text{core})$ similar tokens still fail to reach a sufficient similarity, the using of BCS for token selection will result in information loss and degradation of performance.
> >
> > However, our router can switch from the core-set attention branch to the dense attention branch in case of BCS failure.
> >
> > **End-to-end empirical results show the practical effectiveness of BCS**
> >
> > Furthermore, Figures 8, 12, and 13 in the main manuscript indicate that approximately **40%** of the attention heads are allocated to the core-set attention branch and only **<1%** of the attention heads are allocated to the dense attention.
> >
> > Under this configuration, both the quantitative results (main manuscript Table 1 and 3) and qualitative comparisons (main manuscript Figures 1, 6 and 14) show that the generated videos retain the quality of those produced by the full dense model. These results empirically support the effectiveness of **BCS and router** in enabling **lossless acceleration** for video diffusion models.
> >
> > We are open to incorporating additional validations. If you have specific suggestions for direct metrics or analyses to further test this assumption, we would be glad to include them.
> >
> >
> > ### **Reference**
> >
> > [1] Zhang, et al. “Fast Video Generation with Sliding Tile Attention.” ICML (2025)
> >
> > [2] Xi, et al. “Sparse VideoGen: Accelerating Video Diffusion Transformers with Spatial-Temporal Sparsity.” ICML (2025)
> >
> > [3] Sun, et al. “AsymRnR: Video Diffusion Transformers Acceleration with Asymmetric Reduction and Restoration.” ICML (2025)
> >
> > [4] Bolya, et al. “Token Merging for Fast Stable Diffusion.” CVPR (2023)

---

> > > ### Author Response · Authors · 2025-08-09
> > >
> > > Dear Reviewer m2pg,
> > >
> > > Thank you for engaging in the review and discussion process.
> > >
> > > As a gentle reminder, there are **fewer than 12 hours** remaining in the discussion period. We would appreciate it if you could confirm whether you have any concerns regarding our latest response or any additional points you would like us to address.
> > >
> > > Best regards,
> > > Author

---

> > ### Author Response · Authors · 2025-08-08
> >
> > Dear Reviewer m2pg,
> >
> > Thanks for taking the time to review our work.
> >
> > With the rebuttal deadline approaching in 24 hours, we would like to confirm whether our latest response has addressed your concerns. And do you have any further suggestions?
> >
> > The Authors

---

### Official Review · Reviewer_EhBo · 2025-07-05

**Clarity:** 2
**Significance:** 3
**Originality:** 3
**Rating:** 5
**Confidence:** 1

**Summary:**

VORTA accelerates video diffusion models by dynamically selecting and routing sparse attention mechanisms, improving efficiency with minimal impact on generation quality. It employs coreset selection to reduce attention computation and uses a router to adaptively choose the most suitable attention branch, achieving up to 1.76x end-to-end speedup.

**Questions:**

Please refer to the weakness.

**Ethical Concerns:**

["NO or VERY MINOR ethics concerns only"]

**Final Justification:**

Due to the interest scope and detailed rebuttal provided by author, I will keep my score.

**Limitations:**

yes

**Paper Formatting Concerns:**

No.

**Quality:**

3

**Strengths And Weaknesses:**

**Strength**:
1. The paper addresses a critical issue in video generation, the computational inefficiency of Video Diffusion Transformers. By introducing VORTA, it significantly reduces the sampling time, making VDiTs more practical for real-world applications.
2. The paper presents a novel coreset attention mechanism and a signal-aware routing strategy that dynamically selects the most appropriate sparse attention variant. The bucketed coreset selection (BCS) strategy is an innovative way to handle long-range dependencies efficiently.
3. The experimental results are convincing, demonstrating the effectiveness of VORTA in improving efficiency while maintaining generation quality. The ablation studies provide insights into the importance of each component and the impact of different design choices.

**Weakness**:
1. The demonstration of the proposed corset selection and router strategy is not clear enough, could you please complement more illustration or description?

---

> ### Author Rebuttal · Authors · 2025-07-30
>
> We thank Reviewer **BhBo** for recognizing the practical significance, novelty, and the strength of our experiments and analysis, as well as for the valuable feedback. Below, we address the questions raised.
>
> ---
>
> > **Q1**. The demonstration of the proposed corset selection and router strategy is not clear enough, could you please complement more illustrations or descriptions?
>
> **Bucketed Core-set Selection (BCS) and Core-set Attention** (main manuscript Section 3.2)
>
> 1. Given an input video token sequence $H$, we divide it into non-overlapping spatial-temporal buckets of size $(t, h, w)$. For a concrete example, we set $t=1$, $h=w=3$, and define the core-set retention ratio as $r_\mathrm{core} = \frac{5}{9}$. Each resulting bucket contains $3 \times 3 = 9$ tokens, which we index from 1 to 9 as shown in the layout below.
> | 1 | 2 | 3 |
> | --- | --- | --- |
> | 4 | 5 | 6 |
> | 7 | 8 | 9 |
> 2. Within each bucket, the center token (index 5) is designated as the anchor. This anchor computes the cosine similarity with the remaining 8 tokens in the bucket. The least similar $thw \cdot (1 - r_\mathrm{core}) = 9 \cdot \frac{4}{9} = 4$ tokens are discarded.
> 3. This reduction is motivated by empirical observations that many attention heads in Video DiTs do not rely on fine-grained local details. Thus, semantically redundant tokens can be safely removed without degrading performance.
> 4. After pruning, each bucket retains $r_\mathrm{core}$ of its original tokens, resulting in a compressed sequence termed the ***core-set sequence***.
> 5. Full self-attention is then applied to this shorter sequence, significantly reducing computational cost compared to dense attention.
> 6. Following the attention step, the pruned tokens are restored by assigning them the attention output of the corresponding anchor token (index 5). This reconstructed sequence is then forwarded to subsequent modules, such as the feed-forward layers in the transformer block.
>
> The entire process is performed during inference and is independently recomputed for each attention head.
>
> **Signal-aware Attention Routing** (main manuscript Section 3.3)
>
> 1. For each layer and attention head, a router takes the time-step embedding as input and projects it through a linear layer to a tensor of shape `(batch_size, 3)`. A softmax is then applied, producing three gate values between 0 and 1 that sum to 1.
> 2. **Inference.** Each gate value indicates the relevance of one attention branch for the current head. The branch with the highest gate value is selected for use.
> 3. **Training.** The router is trained to learn suitable branch selection by computing a weighted sum of the outputs from all branches, as defined in Equation 6 of the main manuscript. The gated output is supervised using two losses: one to match the output of dense attention ($\mathcal{L}\_\mathrm{distill}$), and another to align with the ground-truth denoising direction ($\mathcal{L}\_\mathrm{CFM}$).
>
> **A detailed component-wise overhead analysis.** Specifically, we report the latency for a single sampling step (i.e., one forward pass of the model) averaged over 10 prompts and 50 time steps in Table 1 below. All layers and attention heads are configured to use the same attention mechanism for benchmarking. We evaluate three strategies: dense (full) attention, sliding window attention, and core-set attention.
>
> Table 1
>
> |  | Attention | Attention related | Router | BCS | Feed-Forward MLP | Others |
> | --- | --- | --- | --- | --- | --- | --- |
> | Dense | 21197.99 ($\pm 18.52$) | 2912.12 ($\pm 0.74$) | - | - | 269.91 ($\pm 0.14$) | 2.99 ($\pm 0.03$) |
> | Sliding window | 7215.30 ($\pm 14.78$) | 2913.28 ($\pm 0.69$) | 87.23 ($\pm 0.06$) | - | 268.96 ($\pm 0.13$) | 2.98 ($\pm 0.03$) |
> | Core-set | 5647.17 ($\pm 13.40$) | 2911.67 ($\pm 0.72$) | 87.24 ($\pm 0.05$) | 862.73 ($\pm 0.59$) | 267.81 ($\pm 0.11$) | 3.00 ($\pm 0.03$) |
>
> Note:
>
> - All latency values in Table 1 are in milliseconds; values in parentheses denote standard deviation.
> - “Attention-related” includes all operations associated with attention computation except the attention itself (e.g., QKV projections, output projections, and QK normalization).
> - “Others” include operations outside the transformer blocks (e.g., patchification, unpatchification, and ROPE initialization).
> - Latency measurements on HunyuanVideo were performed using a B200 GPU, which was the only available device at the time.
>
> The results indicate that, although the proposed BCS strategy and router module introduce slight additional overhead, this cost is negligible compared to the reduction in attention computation latency.
>
> An overhead analysis of the entire sampling process, conducted on HunyuanVideo with an H100 GPU, is provided in Figure 7 of the main manuscript. And a theoretical complexity analysis of the BCS is detailed in Appendix A.1 of the main manuscript.

---

> > ### Author Response · Authors · 2025-08-06
> >
> > Dear Reviewer EhBo,
> >
> > We sincerely appreciate your taking the time to review our work and your valuable suggestions.
> >
> > With the rebuttal deadline approaching in 2 days, we would like to confirm whether our latest response has addressed your concerns.
> >
> > Thank you once again for your time and effort throughout the review process.
> >
> > The Authors

---

> > ### Comment · Reviewer_EhBo · 2025-08-07
> >
> > I'll keep my score.

---

### Decision · Program_Chairs · 2025-09-17

**Decision:**

Accept (poster)

**Comment:**

This paper proposes VORTA, a routing-based sparse attention framework for accelerating video diffusion transformers. The method achieves substantial speedups with minimal quality loss and integrates well with other efficiency techniques. Reviewers generally praised the novelty, thorough experiments, and practical impact, noting clear ablations and overhead analyses. While one reviewer (m2pg) questioned the empirical validation of the coreset selection assumption and found some explanations overly intuitive, three reviewers recommended acceptance after rebuttal, highlighting the detailed responses and strong experimental evidence. Overall, the paper makes a solid and timely contribution. The meta-reviewer encouages the authors further polish the paper according to the reviewers' comments.